# The importance of interactive chemistry for stratosphere–troposphere coupling

Sabine Haase[1] and Katja Matthes[1,2]

[1]GEOMAR Helmholtz Center for Ocean Research Kiel, Kiel, Germany
[2]Christian-Albrechts-Universität zu Kiel, Kiel, Germany

**Correspondence:** Sabine Haase (shaase@geomar.de)

**Abstract.** Recent observational and modeling studies suggest that not only southern hemispheric surface climate is influenced by stratospheric ozone depletion but also northern hemisphere (NH) spring, implying a strong interaction between dynamics and chemistry. Here, we systematically analyze the importance of interactive chemistry for the representation of stratosphere–troposphere coupling and in particular the effects on NH surface climate during the recent past. We use the interactive and

specified chemistry version of NCAR's Whole Atmosphere Community Climate Model coupled to an ocean model to investigate differences in the mean state of the NH stratosphere as well as in stratospheric extreme events, namely sudden stratospheric warmings (SSWs), and their surface impacts. To be able to focus on differences that arise from two–way interactions between chemistry and dynamics in the model, the specified chemistry model version uses a time–evolving, model–consistent ozone field generated by the interactive chemistry model version. We also test the effects of zonally symmetric versus asymmetric

prescribed ozone, evaluating the importance of ozone waves for the representation of stratospheric mean state and variability. The interactive chemistry simulation is characterized by a statistically significantly stronger and colder polar night jet (PNJ) during spring when ozone depletion becomes important. We identify a negative feedback between lower stratospheric ozone and atmospheric dynamics during the break down of the stratospheric polar vortex in the NH, which contributes to the different characteristics of the PNJ between the simulations. Not only the mean state, but also stratospheric variability is better repre-

sented in the interactive chemistry simulation, which shows a more realistic distribution of SSWs as well as a more persisting surface impact afterwards compared to the simulation where the feedback between chemistry and dynamics is switched off. We hypothesize that this is also related to the feedback between ozone and dynamics through the intrusion of ozone rich air into polar latitudes during SSWs. The results from the zonally asymmetric ozone simulation are closer to the interactive chemistry simulations, implying that under a model–consistent ozone forcing, a three–dimensional representation of the prescribed

ozone field is desirable. This suggests that a 3D ozone forcing as recommended for the upcoming CMIP6 simulations has the potential to improve the representation of stratospheric dynamics and chemistry. Our findings underline the importance of the representation of interactive chemistry and its feedback on the stratospheric mean state and variability not only on the SH but also on the NH during the recent past.

# 1 Introduction

Ozone is a key constituent of the stratosphere and is important not only for stratospheric chemistry, but also for transport and dynamics. Ozone is produced mainly in the tropics and transported towards higher latitudes by the large–scale meridional circulation in the middle atmosphere, i.e. the Brewer Dobson Circulation (BDC). This transport, which is directed towards the winter hemisphere, leads to a larger concentration of ozone at high latitudes compared to lower latitudes. The production of ozone and the absorption of UV radiation by stratospheric ozone leads to the characteristic increase of stratospheric temperature with height resulting in a stable stratification. Hence, ozone and its photochemical characteristics are important for the seasonal cycle of stratospheric temperatures and due to their influence on the meridional temperature gradient also affect stratospheric circulation and dynamics through the thermal wind balance. A large inter–annual variability or anomalous trends in stratospheric ozone have therefore the potential to influence the stratospheric mean dynamical state, its variability as well as stratosphere–troposphere coupling (STC) and, by extension, surface climate. The importance of the interactive representation of stratospheric ozone in a state–of–the–art climate model for STC is addressed here.

It is well known that polar ozone depletion during spring leads to a cooling of the lower stratosphere through radiative heating anomalies (Fig. 1). This cooling in turn enhances catalytic ozone depletion as heterogeneous chemistry is more efficient under lower temperatures (Ⓐ, Fig. 1). It therefore describes a positive feedback based on the interaction between ozone chemistry and absorption of solar radiation (Randel and Wu, 1999). But, there is also a dynamical response to ozone depletion: lower polar temperatures enhance the meridional temperature gradient and hence increase the strength of the polar night jet (PNJ) through thermal wind balance which in turn influences planetary wave propagation and dissipation. Depending on the strength of the PNJ, upward planetary wave propagation and dissipation can either be enhanced or diminished (Charney and Drazin, 1961). This has opposing effects on the state of the polar vortex and can lead either to positive or negative feedbacks between ozone depletion and stratospheric dynamics (Ⓑ and Ⓒ, Fig. 1) (e.g., Mahlman et al., 1994; Manzini et al., 2003; Lin et al., 2017). The strength of the background wind thus determines the impact of ozone depletion on planetary wave propagation and dissipation and hence the sign of the feedback.

If we consider an initial cooling by ozone depletion and strong westerly background winds, this cooling would result in a further strengthening of the background winds which hinders upward planetary wave propagation and hence results in a positive feedback. If the cooling from ozone depletion goes along with weak westerly background winds, this would also result in a strengthening of the background winds but allowing planetary waves to propagate upward and hence resulting in a negative feedback. A stronger (weaker) upward planetary wave propagation results not only in a weakening (strengthening) of the PNJ but also in a strengthening (weakening) of the downwelling branch of the BDC, which can both directly or indirectly influence stratospheric ozone concentrations. A stronger (weaker) descent over the pole leads to an adiabatic warming (cooling) that counteracts (enhances) the negative temperature anomalies induced by ozone depletion (Ⓑ, Fig. 1). A stronger (weaker) descent also increases (decreases) the transport of ozone from higher altitudes to lower altitudes, increasing (decreasing) lower stratospheric ozone concentrations (Ⓒ, Fig. 1). The same effect is achieved by the weaker (stronger) PNJ, which allows for more (less) mixing between ozone depleted polar air masses and relatively ozone rich surrounding air masses. These feedbacks

would therefore be negative (positive) (Ⓑ and Ⓒ, Fig. 1).

The impact of ozone depletion on stratospheric dynamics is strongest during spring (when solar irradiance is available to initiate ozone depletion) and, following our above discussion, will be very sensitive to the background state of the polar vortex. In fact, previous studies suggested a dominance of the negative feedback during the vortex break down (e.g., Manzini et al., 2003; Lin et al., 2017).

Although other trace gases, such as water vapor, can also be affected by these feedbacks, we concentrate our discussion on ozone in this publication. The effects of ozone can be represented differently in climate models: The most sophisticated representation is to calculate ozone interactively within the model's chemistry scheme. Ozone as well as many other trace gases and chemicals is thereby directly and interactively linked to the radiation and dynamics. These climate models are called chemistry–climate models (CCMs) and are used for stratospheric applications such as in the WCRP–SPARC initiatives. However, fully interactive atmospheric chemistry schemes are computationally expensive in particular if also an interactive ocean is used for long–term climate model simulations. An alternative way of representing the effects of ozone chemistry in a climate model is therefore to prescribe ozone fields which can be based on either observed or modeled ozone concentrations. These ozone fields can be of different temporal and horizontal resolution. The majority of climate models that participated in the Climate Model Intercomparison Project, Phase 5, (CMIP5), prescribe ozone as monthly mean, zonal mean values (Eyring et al., 2013) based on the recommended IGAC/SPARC ozone database (Cionni et al., 2011).

When prescribing ozone as monthly mean, zonal mean fields, some aspects of ozone variability, such as zonal asymmetries in ozone, are neglected. Using a monthly climatology was shown to introduce biases in the model's ozone field that reduce the strength of the actual seasonal ozone cycle due to the interpolation of the prescribed ozone field to the model time step (Neely et al., 2014). To avoid these biases, a daily ozone forcing can be applied. Furthermore, ozone is not distributed zonally symmetric in the real atmosphere, therefore prescribing zonal mean ozone values inhibits the effect that zonal asymmetries in ozone, also referred to as ozone waves, can have onto the dynamics. Different studies showed that including zonal asymmetries in ozone in a model simulation would lead to a warmer and weaker stratospheric polar vortex in the NH, which was also associated with a higher frequency in SSWs (e.g., Gabriel et al., 2007; Gillett et al., 2009; McCormack et al., 2011; Peters et al., 2015). The recommended ozone forcing for CMIP6 now includes zonal asymmetries, but does not include variability on time scales smaller than a month (Checa-Garcia et al., 2018).

Since the interactive chemistry module in a climate model is computationally very expensive, it is necessary to elucidate alternative representations of in particular ozone for long–term climate simulations. So far, the importance of interactive chemistry in climate models has been evaluated mainly for experimental settings that focused on the effect of an altered external forcing, such as a change in solar irradiance or in $CO_2$ concentrations (e.g., Chiodo and Polvani, 2016, 2017; Dietmüller et al., 2014; Noda et al., 2018; Nowack et al., 2017, 2018b). In these studies CCM simulations were compared to model simulations forced with a constant ozone field (e.g. based on pre–industrial control conditions), which did not include the ozone response to the changing external forcing. It was shown that the ozone response to the external forcing has an important damping effect onto the surface climate response to the external forcing. Namely, under such conditions, including interactive chemistry reduces the model's climate sensitivity (e.g., Chiodo and Polvani, 2016; Dietmüller et al., 2014; Noda et al., 2018; Nowack et al., 2018b)

and connected surface responses, such as the tropospheric jet (e.g. Chiodo and Polvani, 2017) or ENSO trends (e.g., Nowack et al., 2017). Here, we use a different approach. We are interested in how feedbacks between ozone chemistry and model dynamics can impact the stratospheric mean state and variability given that the variability in stratospheric ozone is the same between the interactive and specified chemistry experiments. This question will be addressed in the present study by using a

time–evolving, model–consistent ozone forcing in the specified chemistry version of the model.

When considering the impact of ozone on stratospheric dynamics one has to distinguish between the two hemispheres. During Antarctic winter, temperatures are very low and reach the threshold for polar stratospheric cloud (PSC) formation every winter. This allows the heterogeneous chemical loss of polar ozone through ozone depleting substances (ODSs) once sunlight returns in spring and leads to the well–known formation of the Antarctic ozone hole every austral spring. Although the Montreal

Protocol regulated the emissions of ODSs, they have a very long life–time and continue to deplete ozone every winter, most prominently seen in the last two decades of the $20^{th}$ century. The ozone hole contributed to a positive trend in the southern annular mode during austral summer (December to February, DJF), which influences the position and strength of the tropospheric jet and thereby impacts the surface wind stress forcing on the Southern Ocean (e.g., Son et al., 2008; Thompson et al., 2011; Previdi and Polvani, 2014).

Recently Son et al. (2018) evaluated the representation of the observed SH ozone trend and the resulting poleward shift of the tropospheric jet in the latest CCMs and high–top CMIP5 models (model top at or above 1 hPa). They argue that irrespective of the representation of stratospheric ozone (prescribed or interactive) the poleward shift of the tropospheric jet due to ozone depletion was captured in all model ensembles. Separating those CMIP5 models with and without interactive chemistry showed a slightly stronger poleward trend in zonal mean zonal wind during DJF in the models with interactive chemistry. However,

Son et al. (2018) also point out that the inter model spread in tropospheric jet latitude trend is rather high. It is positively correlated to the strength of the ozone trend in individual CCMs but also dependent on different model dynamics. It is therefore more convenient to use one model with the same dynamics to investigate the effect of interactive chemistry. For example, Li et al. (2016) focused on one model, the Goddard Earth Observing System Model version 5 (GEOS–5), to assure for the same dynamical background between simulations and found a significantly stronger trend in zonal mean zonal wind in austral

summer and a more significant surface response in surface wind stress and ocean circulation to the same ozone trends when ozone was calculated interactively in the model. There are only a few studies, like that of Li et al. (2016), that are designed to systematically compare the effect of including or excluding interactive chemistry in the same model, i.e. using the ozone forcing from the CCM also in the specified chemistry version of the model. But there is still a great need to better understand the role that feedbacks between chemistry and dynamics may play in representing recent and also future climate conditions on

different time scales.

Recently, Lin et al. (2017) discussed the negative feedback between ozone depletion and dynamics (recall Fig. 1) in detail for the observed SH ozone trend showing that the lower stratospheric dynamical response to ozone depletion depends on the timing of the climatological vortex break down during spring. They also claim that models with a cold pole bias overestimate the effect of SH ozone depletion due to an underestimation of the negative feedback. Here, we want to investigate how important

the representation of such feedbacks in a climate model is for northern hemisphere (NH) stratospheric dynamics and whether

it can impact the tropospheric circulation via extreme stratospheric events.

On the NH, where the stratospheric polar vortex is much more disturbed and therefore warmer during winter, a clear trend in either total column or lower stratospheric ozone is not as prominent as in the SH. Very low ozone concentrations dominated in the 1990s (Ivy et al., 2017), but also more recent years, such as 2011, reached extremely low Arctic spring ozone concen-

trations (Manney et al., 2011). This event in particular initiated discussions about the possibility of an Arctic ozone hole and also on a possible impact of NH ozone depletion events on the surface (Cheung et al., 2014; Karpechko et al., 2014; Smith and Polvani, 2014). Using different models but all with prescribed ozone, these studies did not find a significant surface impact from observed ozone anomalies. In particular, Smith and Polvani (2014) reported that significantly larger NH ozone depletion than that observed in 2011 would be needed for a detectable surface impact. On the other hand, Calvo et al. (2015) report about

statistically significant impacts of NH ozone depletion events on tropospheric winds, surface temperatures and precipitation in April and May using the same CCM (WACCM) as used in this study. This suggests that feedbacks between dynamics and chemistry are necessary to induce a tropospheric signal due to ozone depletion on the NH. We will test the importance of two–way feedbacks between ozone chemistry and dynamics for NH STC in recent decades here.

Extreme events in the NH stratosphere can have strong and relatively long–lasting impacts on the troposphere (e.g. Baldwin

and Dunkerton, 2001) and are therefore of great interest, for example, for seasonal weather prediction (e.g. Baldwin et al., 2003; Sigmond et al., 2013). Different pathways have been proposed to explain the coupling between the stratosphere and the troposphere, including wave–mean flow interaction, wave refraction and reflection mechanisms (e.g., Haynes et al., 1991; Hartmann et al., 2000; Perlwitz and Harnik, 2003; Song and Robinson, 2004) as well as potential vorticity change (Ambaum and Hoskins, 2002; Black, 2002). Understanding the relative contribution of these mechanisms to STC in detail is still subject

of recent research. Here, we focus on sudden stratospheric warmings (SSWs) as a prominent example of NH STC. SSWs are characterized by a strong wave–driven disturbance or break–down of the stratospheric polar vortex and result in a surface response a few days after the onset of the stratospheric event that resembles the pattern of the negative phase of the North Atlantic Oscillation (NAO) (Baldwin and Dunkerton, 2001). A systematic investigation of interactive vs. prescribed ozone in the same climate model family on NH STC effects has to our knowledge not yet been performed and is the goal of the present

study.

Apart from the representation of two–way feedbacks between chemistry and dynamics, also zonal asymmetry in ozone is often not included when ozone and other radiatively active species are prescribed. But, earlier publications showed that zonally asymmetric ozone is associated with a warmer and weaker stratospheric polar vortex in the NH (e.g. Gillett et al., 2009; McCormack et al., 2011; Albers and Nathan, 2012; Peters et al., 2015) compared to zonal mean ozone conditions. Gillett et al.

(2009), for example, showed that the NH polar stratospheric vortex is warmer when using zonally asymmetric ozone rather than zonal mean ozone in the radiation scheme. In their model setup feedbacks between dynamics and zonal mean ozone concentrations are possible, only the effects of ozone waves are inhibited. A significant warming of the polar stratosphere was found only in early winter (November and December). Using a similar model setup, McCormack et al. (2011) found a more significant warming in February when including zonally asymmetric ozone in their model and connected it to the higher

abundance of SSWs in their experiments. The total number of SSWs was rather low with only 5 out of 30 ensemble members.

out of 5 SSWs occurred in the zonally asymmetric simulations. Peters et al. (2015) prescribed ozone in both simulations and also found a larger abundance of SSWs in the zonally asymmetric ozone run with the largest difference in SSW occurrence in November. Furthermore, a recent study by Silverman et al. (2018) points to the importance of the Quasi-Biennial Oscillation (QBO) for the NH high latitude response to ozone waves. To test the sensitivity of using either a zonal mean ozone field or a zonally asymmetric one, we additionally include a sensitivity experiment using a 3D ozone forcing in the specified chemistry simulation.

The paper is organized as follows: Section 2 introduces the model and the simulations performed in this study together with the applied methodologies. After discussing the differences in the climatological mean state between interactive and prescribed chemistry model simulations in section 3, we analyze the differences in SSW characteristics and downward influences between the simulations in section 4. We conclude the paper with a discussion of our results.

## 2 Data and Methods

### 2.1 Model Simulations

To asses the importance of interactive chemistry on the mean state and variability of the stratosphere as well as on STC, we use a model that is capable of using an interactive chemistry scheme as well specified chemistry.

We use the Community Earth System Model (CESM), version 1, from NCAR with WACCM, version 4, as the atmospheric component; this setting is referred to as CESM1(WACCM). This version of CESM1(WACCM) has been documented in detail in Marsh et al. (2013).

WACCM is a fully interactive chemistry–climate model, with a horizontal resolution of 1.9°latitude by 2.5°longitude. It uses a finite volume dynamical core, has 66 vertical levels with variable spacing and an upper lid at $5.1x10^{-6}$hPa (about $140\,\mathrm{km}$) that reaches into the lower thermosphere (Garcia et al., 2007). Stratospheric variability, such as SSW properties and the evolution of the SH ozone hole are well captured in CESM1(WACCM) (Marsh et al., 2013). On the SH, CESM1(WACCM) has a strong cold pole bias in the middle atmosphere, which could influence the feedbacks discussed in Figure 1 (Lin et al., 2017). On the NH, the strength of the PNJ agrees well with observations (Richter et al., 2010) and therefore the NH is better suited to investigate these feedbacks.

For our investigations we run the model under historical forcing conditions for the period of 1955 to 2005 and under the representative concentration pathway 8.5 (RCP8.5) from 2006 to 2019. We thereby capture a 65–year period that features the years with lowest ozone concentrations before ozone recovery starts. We include all external forcings based on the CMIP5 recommendations: GHG and ODS concentrations (Meinshausen et al., 2011), spectral solar irradiances (Lean et al., 2005), and volcanic aerosol concentrations (Tilmes et al., 2009) including the eruptions of Agung (1963), El Chichón (1982), and Mount Pinatubo (1991). As the Quasi-Biennial Oscillation (QBO) is not generated internally by this version of WACCM, the QBO was nudged following the methodology of Matthes et al. (2010).

CESM1(WACCM) incorporates an active ocean (Parallel Ocean Program version 2, POP2), land (Community Land Model

version 4, CLM4) and sea ice (Community Ice CodE version 4, CICE4) model. POP2 and CICE4 have a nominal latitude-longitude resolution of 1°; the ocean model has 60 vertical levels. A central coupler is used to exchange fluxes between the different components. For more details on the different model components the reader is referred to Hurrell et al. (2013) and references therein.

As mentioned above, WACCM incorporates an interactive chemistry scheme in its standard version. It uses version 3 of the Model for Ozone and Related Chemical Tracers (MOZART) (Kinnison et al., 2007). Within MOZART ozone concentrations and concentrations of other radiatively active species are calculated interactively, which allows for feedbacks between dynamics and chemistry as well as radiation. It includes the $O_X$, $NO_X$, $HO_X$, $ClO_X$, and $BrO_X$ chemical families, along with $CH_4$ and its degradation products. A total of 59 species and 217 gas phase chemical reactions are represented and 17 heterogeneous
reactions on three aerosol types are included (Kinnison et al., 2007).

The specified chemistry version of WACCM (SC-WACCM), in which interactive chemistry is turned off, does not simulate feedbacks between chemistry and dynamics. This version of WACCM is documented in Smith et al. (2014). Here, ozone concentrations are prescribed throughout the whole atmosphere. Above approximately 65 km additionally to the ozone concentrations, also concentrations of other species, namely atomic and molecular oxygen, carbon dioxide, nitrogen oxide and
hydrogen, as well as the total shortwave and chemical heating rates are prescribed. Smith et al. (2014) validated SC-WACCM with prescribing monthly mean zonal mean values of the aforementioned species and heating rates from a companion WACCM run. Following the procedure in Smith et al. (2014) we use the output from our transient WACCM integration to specify all necessary components in SC-WACCM (i.e. O, $O_2$, $O_3$, NO, H, $CO_2$ and total short–wave and chemical heating rates). We use transient, monthly mean zonal mean values for all variables, except ozone, for which we use daily zonal mean transient data.
The use of daily ozone data reduces a bias that is introduced by linear interpolation of the prescribed ozone data to the model time step when using monthly ozone values (Neely et al., 2014). Using daily data also allows for extreme ozone anomalies to occur in the specified chemistry run.

In the following we will refer to the interactive chemistry version of CESM1(WACCM) as "Chem ON" and to the specified version, that uses SC-WACCM as the atmosphere component, as "Chem OFF". Additionally, we include results from a sensi-
tivity run, prescribing daily zonally asymmetric (3D) transient ozone in SC-WACCM, which will be referred to as Chem OFF 3D. All other settings in Chem OFF 3D are equal to that of the Chem OFF simulation. The model simulations and settings are summarized in Table 1.

## 2.2   Methods

The results presented in this paper are largely based on climatological mean values of model output. When variability is considered we use deseasonalized daily or monthly data by removing a slowly varying climatology after removing the global mean from each grid point each year. This follows the procedure described in Gerber et al. (2010) and is used to omit the effect that may arise from variability on timescales larger than 30 years, such as the signature of global warming. The slowly varying climatology is produced as follows: First, a 60 day low pass filter is applied. Then, for each time step and grid point, a 30

year low pass filter is applied to the smoothed time series. Gerber et al. (2010) describe this procedure in detail and apply it exemplarily. We confine the presented results to altitudes below 5 hPa since it is the lower stratospheric ozone and its effects on the circulation that we are most interested in.

We calculated the vertical component of the meridional residual circulation ($\overline{w}^*$) using the transformed Eulerian mean framework defined for example in Andrews et al. (1987):

$$\overline{w}^* = \overline{w} + \frac{1}{A cos\phi}\left(cos\phi \frac{\overline{v'\Theta'}}{\overline{\Theta}'_z}\right)_\phi.$$

With the overbar indicating zonal mean values and subscripts referring to partial derivatives. A denotes the Earth's radius (A = 6371000 m). $\overline{w}^*$ is used to estimate the difference in tropical upwelling and polar downwelling between the model simulations. We will refer to major sudden stratospheric warmings simply as "SSWs" or "major warmings" in the following. SSWs are defined based on the definition of the World Meteorological Organisation (WMO) (e.g., McInturff, 1978; Andrews et al., 1987), after which they occur (between November and March) when two criteria are fulfilled: 1) the predominantly westerly zonal mean zonal wind reverses sign at 60°N and 10 hPa, i.e. changes from westerly to easterly; and 2) the 10 hPa zonal mean temperature difference between 60°N and the pole is positive for at least 5 consecutive days. The central date (or onset) of SSWs is defined as the first day of wind reversal. To exclude final warmings (the transition from winter to summer circulation), a switch from westerly to easterly winds at the given location is only considered a SSW if the westerly wind recovers for at least 10 consecutive days prior to April $30^{th}$ (Charlton and Polvani, 2007) and exceeds a threshold of 5 ms$^{-1}$ (Bancalá et al., 2012). To avoid double counting of events, there have to be at least 20 days of westerlies in between two major warmings (Charlton and Polvani, 2007).

We compare the modeled major warming frequency to the European Centre for Medium-Range Weather Forecasts Re-Analysis (ERA) products ERA40 (Uppala et al., 2005) and ERA-Interim (Dee et al., 2011). These two products were combined into one data set following Blume et al. (2012) (here merged on the 1st of April 1979), which resolves the stratosphere up to 1 hPa and spans the period from 1958 to 2017.

Regarding the uncertainty estimate for the SSW frequencies we use the standard error for the monthly frequencies and the 95% confidence interval based on the standard error for the winter mean frequency.

Atmospheric variability linked to SSWs is evaluated in the form of composites for selected variables before, during and after the SSW onset. Statistical significance of the composites is tested using a Monte Carlo approach (see for example von Storch and Zwiers, 1999). Therefore, 10000 randomly chosen central dates are used to calculate random composites. Statistical significance at the 95% level is reached when the actual composites exceed the $2.5^{th}$ or $97.5^{th}$ percentiles of the distribution drawn from the random composites.

The differences between Chem ON and Chem OFF are displayed as the difference: Chem ON minus Chem OFF and are depicted together with the climatological field of the Chem OFF run to display the effect of including interactive chemistry. For these differences, statistical significance at the 90% or 95% level is tested using a two-sided t-test.

## 3 The impact of interactive chemistry on the stratospheric mean state

To assess the importance of interactive chemistry on stratospheric dynamics we first consider zonal mean zonal wind at 10 hPa (U10) and zonal mean temperature at 30 hPa (T30) to characterize the stratospheric polar vortex in our model simulations (Figs. 2a and b). The stratospheric PNJ is characterized by strong westerlies around 70°N and 60°S (Fig. 2a) and low polar cap
temperatures (Fig. 2b). The PNJ is significantly stronger and colder in the Chem ON run. On both hemispheres, this feature is especially significant during spring, when ozone chemistry becomes important for the temperature budget of the lower stratosphere and hence for the dynamics. This difference already hints at the relevance of representing feedbacks between ozone chemistry and dynamics for the climatological state of the PNJ during spring. On the NH, the difference between the runs is also significant during fall and early winter, which is connected to a weaker downwelling, i.e. weaker adiabatic warming,
indicated by the statistically significant positive anomaly in $\overline{w}^*$ at 70 hPa (Fig. 2c) from June to December. At the same time Chem ON is characterized by a slightly weaker tropical upwelling at 70 hPa, indicating that at least the shallow branch of the BDC (below 50 hPa) is weaker in Chem ON compared to Chem OFF.

In the following we will focus on the NH spring season as this is the period when the effect of ozone depletion and possible feedbacks between chemistry and dynamics become important. Figure 3 shows February to April (FMA) NH zonal mean zonal
wind and zonal mean temperature with height. Consistent with Figure 2, north of 70°N, we find a stronger PNJ (up to 4.5 ms$^{-1}$ stronger at about 10 hPa) when interactive chemistry is included (Fig. 3a) and a colder polar vortex, with a maximum difference between Chem ON and Chem OFF of -2.8 K at about 60 hPa directly at the pole (Fig. 3b). While temperature differences between Chem ON and Chem OFF are mainly restricted to the lower stratosphere, statistically significant differences in zonal mean zonal wind reach up to about 4 hPa and even down to the surface.

As the temperature differences are decisive for the differences in zonal wind, we now consider the differences in polar cap heating rates between Chem ON and Chem OFF to investigate why the models differ in their climatological stratospheric state (Fig. 4). As already seen in Figures 2 and 3, including interactive chemistry leads to a stronger PNJ and colder polar vortex, especially during spring but also during early winter (Figure 4a and b). Figures 4a and c show that lower (higher) temperatures go along with weaker (stronger) long–wave (LW) cooling in the Chem ON run. The difference in LW cooling between Chem
ON and Chem OFF is directly connected to the temperature difference and works as a damping factor. By construction, there are no significant differences in the short–wave (SW) heating rates between Chem ON and Chem OFF that could explain the different temperatures between the models in this region, neither can differences in temperature tendencies due to gravity waves (not shown). The dynamical heating rates, which describe the total adiabatic heating rates in the model dominated by advection through the vertical component of the residual circulation, $(\overline{w}^*)$, (Fig. 4d) seem to be the dominant factor in shap-
ing the climatological differences in polar cap temperature between Chem ON and Chem OFF. Although the spring season is characterized by a stronger PNJ and lower polar cap temperatures in the lower stratosphere in Chem ON, a stronger dynamical heating in April and May leads to higher temperatures in Chem ON in the middle stratosphere peaking in May (Fig. 4a and d). Statistically significant dynamical heating differences between Chem ON and Chem OFF reach down to the troposphere resulting in a strong reduction of the temperature difference between Chem ON and Chem OFF in the lower stratosphere in

May. These features are characteristic for a later but more intense break down of the polar vortex when interactive chemistry is present. The differences in temperature between Chem ON and Chem OFF during early winter can be explained by the differences in dynamical heating as well. In the Chem ON run there is statistically significant weaker dynamical warming as compared to the Chem OFF run with a maximum difference between the runs in November (Fig. 4d) that leads to lower

temperatures in Chem ON in December. This agrees with the earlier finding that the shallow branch of the BDC is weaker in the Chem ON simulation (Fig. 2c). Why does the signal in dynamical heating differ between early winter and late spring? We suggest feedbacks between ozone chemistry and dynamics to be the reason for that and will discuss this in more detail in the following.

To illustrate the relation between ozone and dynamical heating we calculated the correlation between polar cap ozone con-

centrations at 50 hPa and polar cap dynamical heating rates in Chem ON and Chem OFF. A similar analysis using ozone and temperature was carried out by Lin et al. (2017) for the SH. Figure 5 shows this correlation for ozone lagging and leading the dynamical heating rates by 15 days. As the dynamical heating is only available in monthly resolution, daily ozone data was shifted by -/+ 15 days with respect to the dynamical heating time axis. The contours show the climatological zonal mean zonal wind as a reference. The shading shows the correlation coefficients. Two different states are represented in Figure 5: 1) the

dependence of ozone on the dynamics (Fig. 5, top row) and 2) the effect ozone can have on the dynamics (Fig. 5, bottom row). When ozone lags behind dynamical heating (Fig. 5a, top row), positive correlation coefficients occur in late autumn and early winter indicating that low (high) ozone concentrations follow low (high) dynamical heating rates. In this case, ozone concentrations and dynamical heating are caused by a reduced (enhanced) downwelling which leads to adiabatic cooling (warming) as well as to lower (higher) ozone concentrations. When ozone leads dynamical heating (Fig. 5a, bottom row), positive correlation

coefficients are not significant anymore. Instead, a statistically significant negative correlation between ozone and dynamical heating throughout the lower stratosphere is found in April and May, setting in earlier at higher altitudes (above 10 hPa). By only looking at the dynamical heating rates here, we do not capture possible positive feedbacks caused by radiative heating and ozone chemistry indicated under Ⓐ in Figure 1. Using this analysis we also do not identify a positive feedback between ozone chemistry and dynamics (recall Fig. (Ⓑ and Ⓒ, Fig. 1)). But, we clearly find a negative feedback between ozone

and dynamics during the vortex break down phase in correspondence to earlier studies (e.g. Manzini et al., 2003; Lin et al., 2017). The westerly background wind is sufficiently weak so that a decrease in ozone concentrations leads to an increase in dynamical heating, which would in turn increase ozone concentrations via the aforementioned pathways (Ⓑ and Ⓒ, Fig. 1). This negative feedback indicates that during weak zonal mean zonal wind conditions, ozone depletion, which leads to an initial cooling of the lower polar stratosphere and strengthening of the PNJ, eventually leads to a faster break down of the vortex by

allowing upward wave propagation to take place at a higher rate than it would be during weaker westerlies. In this analysis, the negative feedback clearly dominates and leads to a more abrupt break–down of the polar vortex in the Chem ON simulation. A statistically significant correlation signature between ozone and dynamical heating is only found in Chem ON (compare Figs. 5a and b). Hence, we conclude that interactive chemistry is indeed contributing to the different climatological characteristics of the PNJ between Chem ON and Chem OFF.

Apart from the lack of feedbacks between chemistry and dynamics, Chem OFF is also missing zonal asymmetry in the pre-

scribed ozone field. Hence, the missing effect of ozone waves in the Chem OFF simulation can potentially contribute to the differences that we find between Chem ON and Chem OFF. We therefore also include a sensitivity run, for that we used a zonally asymmetric daily ozone forcing, Chem OFF 3D (Table 1).

When including ozone waves, there is, similarly to Chem OFF, no significant correlation signature found between ozone and dynamical heating (not shown). Nevertheless, the absolute climatological differences between Chem ON and Chem OFF 3D are smaller compared to what we found for a zonally symmetric ozone forcing (Figs. 4 and 6). The PNJ is still colder and stronger with interactive chemistry (Figs. 6a and b) and significant differences of the same sign as above are found for LW and dynamical heating rates in the spring season (Figs. 6c and d). The lower amplitude of the differences between Chem ON and Chem OFF 3D as compared to Chem ON and Chem OFF do indicate that also other processes (apart from the feedbacks discussed so far) are important for the generally stronger and colder PNJ in Chem ON. Including zonal asymmetries in ozone does allow for stronger anomalies in ozone in general since no averaging is applied and for anomalies that do not center over the pole but affect lower latitudes as well. Hence, ozone waves can influence wave propagation and dissipation pathways possibly leading to a better representation of the effect that ozone has on wave–mean flow interactions in our model setup.

## 4   How does interactive chemistry influence stratosphere–troposphere coupling?

We found a stronger PNJ during NH spring when interactive chemistry and feedbacks between ozone and dynamics are included in a climate model. This stronger PNJ exhibits a boundary for upward planetary wave propagation which influences the occurrence of SSWs. Figure 7 shows the frequency of SSWs for ERA reanalysis data (gray), the Chem ON (blue), Chem OFF (light green) and Chem OFF 3D (dark green) simulations for each month of the extended winter season individually (left) and the average over the whole winter season (right) (see also Table 1). Chem ON represents the observed monthly frequency of SSWs very well with the exception of January where it significantly underestimates the occurrence of SSWs. Chem OFF on the other hand underestimates SSWs significantly in February and shows an unrealistic increase in occurrence of SSWs towards the end of the extended winter season (March). Overall there is a tendency for fewer SSWs when interactive chemistry is included in the model (Chem ON: 0.41 +- 0.12 warmings per winter, Chem OFF: 0.64 +- 0.12 warmings per winter, and Table 1), which is likely due to the stronger background westerlies in Chem ON. The SSW frequency in Chem OFF 3D is much closer to that in Chem ON as compared to Chem OFF, which we attribute to the smaller climatological differences between Chem ON and Chem OFF 3D. But how does interactive chemistry impact the downward influence of SSWs?

The downward propagation of anomalies connected to the vortex break down is stronger in the Chem ON simulation (Fig. 8). Polar cap temperature anomalies are stronger and persist longer in Chem ON (Fig. 8a). Also the zonal mean wind at 60°N (Fig. 8b) shows a longer lasting easterly anomaly connected to SSWs that reaches further down to the surface. Figures 8a and b also demonstrate that the SSW signal in the Chem ON run is more sudden compared to the Chem OFF run: the polar cap temperature anomaly is significantly weaker before and significantly stronger after the SSW onset compared to the Chem OFF run. Also, the easterly wind at 60°N is preceded by stronger westerlies in the Chem ON simulation. Both criteria show a more

abrupt change from before to after the central date. To consider the possible impact of ozone chemistry, we additionally show a composite of ozone volume mixing ratio anomalies during the SSWs (Fig. 8c). A strong intrusion of ozone from surrounding air masses during the SSWs, as described in de la Cámara et al. (2018), is evident only in the Chem ON simulation. No significant signal is found in the Chem OFF run (contours in Fig. 8c). This suggests that the increase in lower stratospheric ozone in

Chem ON contributes to the longer persistence of the SSW signal in the lower stratosphere.

The stronger and more persistent SSW signal in the Chem ON run in the stratosphere appears also at the surface in the sea level pressure (SLP) response to SSWs (Fig. 9). The well known negative NAO–like surface response after SSWs is stronger in the Chem ON simulation (averaged over 30 days after the SSW onset, Fig. 9a) and longer lasting (averaged over 30 to 60 days after the SSW onset, Fig. 9e) compared to the Chem OFF simulation (Figs. 9b and f). This larger persistence of SLP anomalies

after SSWs, which we also find in the combined ERA data set (Figs. 9d and h), could be due to the intrusion of ozone into the lower stratosphere that is represented only with interactive chemistry (Fig. 8c). Prescribing zonally asymmetric ozone does not significantly improve the surface response (Figs. 9c and g). The NAO signal averaged over 30 days after the SSWs is similar to Chem OFF, and restricted to a significant positive anomaly over the pole 30 to 60 days after the SSW. Hence, a prescribed 3D ozone forcing is not sufficient to simulate the persistent NAO–like SLP signal after SSWs.

## 5   Conclusions

In this study we systematically investigated the effect of interactive chemistry on the characteristics of the stratospheric polar vortex in CESM1(WACCM) during the second half of the $20^{th}$ century and the beginning of the $21^{st}$ century with a focus on the NH climatology as well as on its interannual variability. Therefore, an interactive chemistry climate model was compared

to the specified chemistry version of the same model using a time–evolving, model–consistent, daily ozone forcing. We found that including interactive chemistry (Chem ON) results in a colder and stronger polar night jet (PNJ) during spring and early winter. We attribute the spring difference to feedbacks between the model dynamics and ozone chemistry (Fig. 1). The inability to include a dynamically consistent ozone variability when prescribing ozone (Chem OFF), inhibits the two–way interaction between ozone chemistry and model dynamics. We found a negative feedback between ozone chemistry and dynamics similar

to that described by Lin et al. (2017) for the SH to be very important during the break down of the NH polar vortex in our Chem ON simulation: An initial polar cap temperature decrease due to ozone depletion during NH spring occurs in correspondence with an increase in the strength of the PNJ, which during weak background westerlies leads to an increase in upward planetary wave propagation and dissipation and hence results in adiabatic warming and increase in ozone due to a stronger descent of air masses. This negative feedback, which only appears in the Chem ON simulation (Fig. 5), leads to a more abrupt transition from

the winter to the summer circulation. The climatological differences between Chem ON and Chem OFF during early winter result from reduced dynamical heating in the Chem ON simulation, associated with a weaker polar downwelling (Fig. 2c and Fig. 4d).

The climatological differences between the model simulations also influence stratosphere–troposphere coupling. The distribu-

tion of SSWs is very well captured in Chem ON, while Chem OFF significantly overestimates SSWs in March, when ozone chemistry is most important (Fig. 7). The stratospheric anomalies in polar cap temperature and mid latitude zonal wind associated with SSWs as well as the NAO–like SLP response to SSWs are better captured and longer persistent in the Chem ON simulation (Figs. 8 and 9). Hence, feedbacks between chemistry and dynamics may also impact the influence that stratospheric

events can have on the troposphere. In Chem ON, ozone rich air from surrounding air masses is mixed into the polar vortex during SSWs in correspondence to de la Cámara et al. (2018). Additional heating due to the increase in ozone mixing ratios could explain the extended lifetime of the SSW warming signal in the lower stratosphere in Chem ON and thereby the longer persistence of the NAO–like SLP anomaly in association with the occurrence of SSWs in the Chem ON simulation.

Apart from the lack of feedbacks between chemistry and dynamics, Chem OFF is also missing the effect of ozone waves in

the prescribed zonal mean ozone field, which contributes to the differences between Chem ON and Chem OFF. We therefore performed a sensitivity run prescribing zonally asymmetric (3D) ozone (Chem OFF 3D, Table 1). The differences between Chem ON and Chem OFF 3D agree in sign to that of the differences between Chem ON and Chem OFF but are overall smaller in amplitude and less significant (Figs. 4 and 6). Significant differences are restricted to early winter and late spring. We hence conclude that the missing effects of ozone waves in Chem OFF are contributing to the larger differences between Chem ON

and Chem OFF.

Considering stratospheric variability, the distribution of SSWs throughout the winter season is still better captured in Chem ON compared to Chem OFF 3D (Fig. 7), whereas the total SSW frequency in Chem OFF 3D is not significantly different from that in Chem ON (Table 1). Also, the SSW surface impact is better captured in Chem ON as compared to Chem OFF 3D (Fig. 9), which we explain with the missing intrusion of ozone rich air into higher latitudes in Chem OFF 3D (similar to Chem OFF)

(not shown).

Our results demonstrate the importance of chemistry–dynamics–interactions and also hint to an important influence of ozone waves on the differences between Chem ON and Chem OFF. Prescribing daily zonally asymmetric ozone such as in Chem OFF 3D, which is not consistent with the dynamics might also introduce feedbacks that are difficult to interpret. A larger ensemble of experiments, which was unfortunately not possible for this study, is needed to better understand the importance of feedbacks

between chemistry and dynamics in the absence and presence of ozone waves. Therefore, a larger ensemble of simulations is planned for a follow–up study to increase significance and reduce the effect of internal variability on the results. However, to further validate the results presented in this study, we show the difference in the climatological mean state of the middle stratosphere for a 145 year control simulation in Figure 10 using a constant external forcing based on 1960s conditions. Zonal wind and temperature show the same differences between Chem ON CTRL and Chem OFF CTRL as presented in Fig. 2 for the

transient forcing. The amplitude of the differences is lower, which we attribute to the lower variability in lower stratospheric ozone in this control setting. It nevertheless shows that our basic results are robust and can be reproduced in a control setting.

It is however essential to better understand the role of chemistry–dynamics–interactions in order to improve our decisions about how ozone shall be prescribed in upcoming model simulations. A new approach was recently presented by Nowack et al. (2018a), who discuss the potential of machine learning to parameterize the impact of ozone in different standard scenarios,

such as in a $4xCO_2$ setting. Based on our findings from prescribing a model–consistent, daily ozone forcing, we argue that a

3D ozone forcing as now provided for CMIP6 has the potential to improve the representation of the impact that ozone chemistry has on model dynamics. However, such a forcing does not perfectly compare to our experimental setting since the more generalized CMIP6 ozone forcing cannot supply model–consistent ozone fields for different models and is based on monthly mean data.

*Data availability.* Reanalysis data used in this paper is publicly available from the ECMWF for the ERA-40 and ERA-Interim products. CESM1(WACCM) model data requests should be addressed to Katja Matthes (kmatthes@geomar.de).

*Author contributions.* SH and KM designed the model experiments, decided about the analysis and wrote the paper. SH carried out the model simulations and data analysis and produced all the figures.

*Competing interests.* The authors declare that they have no competing interests.

*Acknowledgements.* We want to thank two anonymous reviewers for their constructive comments that helped to improve the manuscript. We thank the computing center at Christian–Albrechts–University in Kiel for support and computer time.

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

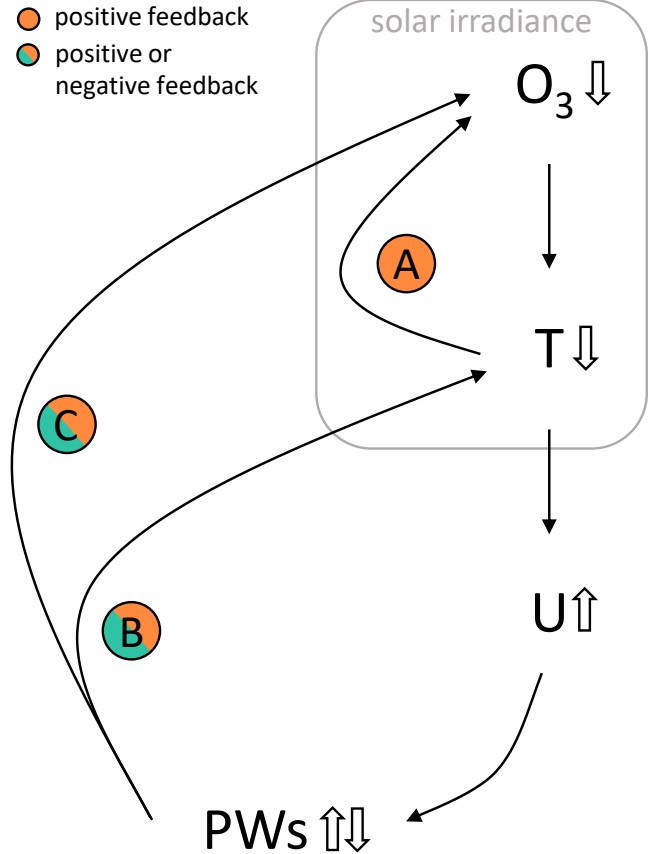

**Figure 1.** Scheme of possible feedbacks between ozone chemistry and dynamics/transport. A negative anomaly in ozone (O3) will lead to a negative anomaly in temperature (T) which favors ozone depletion (A, positive feedback). It also increases the strength of the polar night jet (U). Depending on the strength of the background westerlies an increase in U can lead to either an increase or decrease in upward planetary wave propagation (PWs). A strong (weak) westerly background wind would lead to a decrease (increase) in PWs, which is connected to a less (more) disturbed polar vortex, connected to (B) a cooling (warming) of the polar vortex and (C) to less (more) transport of ozone into the polar vortex. Strong (weak) background westerlies are therefore connected to positive (negative) feedbacks between ozone chemistry and dynamics/transport (B and C).

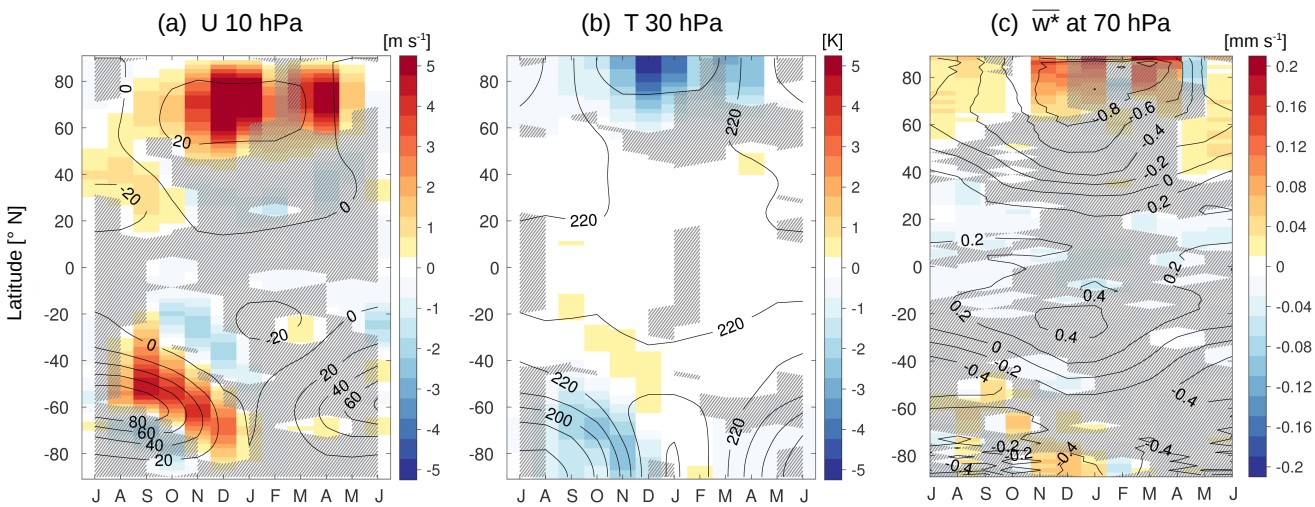

**Figure 2.** Climatological zonal mean a) zonal wind at 10 hPa in $\mathrm{ms}^{-1}$, b) temperature at 30 hPa in K and c) $\overline{w}^*$ at 70 hPa in $\mathrm{mms}^{-1}$ with month and latitude for Chem OFF (contours) and for the difference between Chem ON and Chem OFF (shading). Contour intervals are a) $20\,\mathrm{ms}^{-1}$, b) 10 K, and c) $0.2\,\mathrm{mms}^{-1}$. Statistically insignificant areas are hatched at the 95% level.

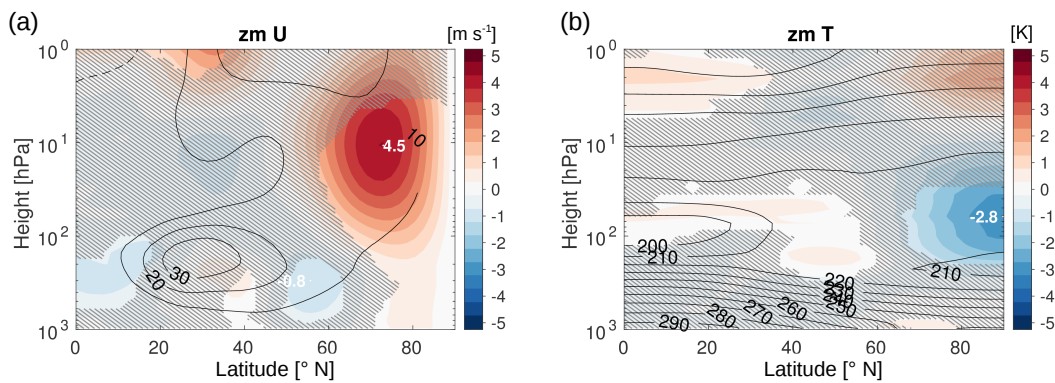

**Figure 3.** FMA zonal mean a) zonal wind in $ms^{-1}$, b) temperature in K with latitude and height for the NH for Chem OFF (contours) and for the difference between Chem ON and Chem OFF (shading). Contour intervals are a) $10\ ms^{-1}$, and b) 10 K. Solid contours are used for positive values, dashed contours are used for negative values. The zero line is omitted. Statistically insignificant areas are hatched at the 95% level.

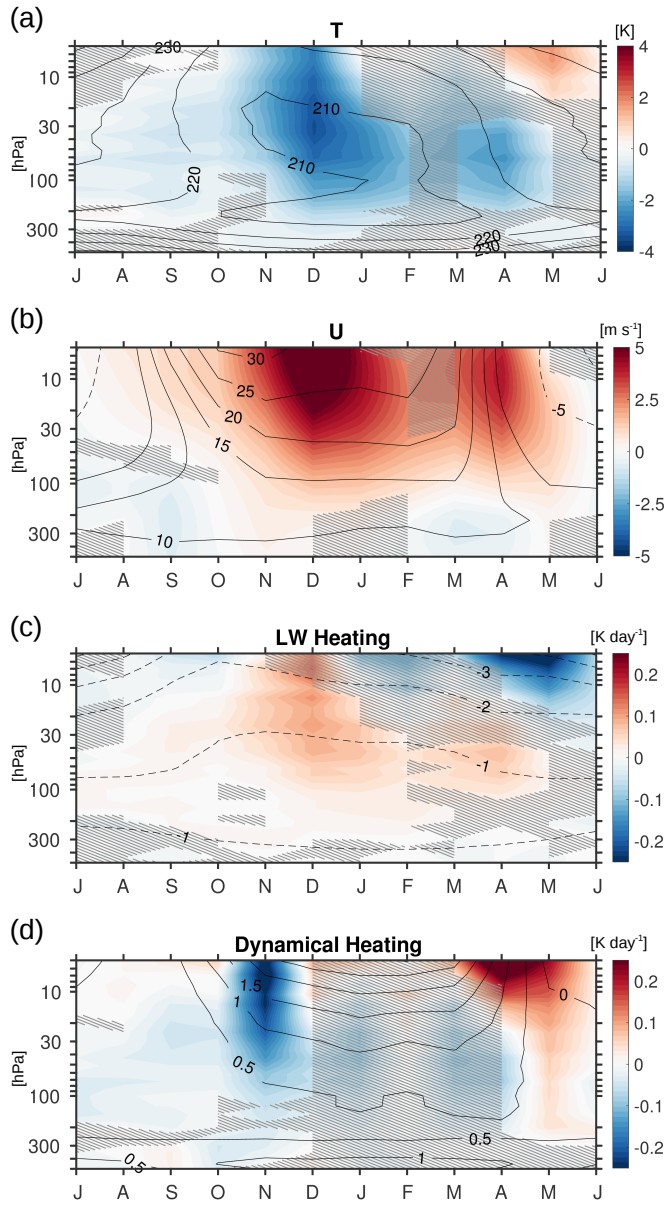

**Figure 4.** Climatological NH a) polar cap ($70°$ to $90°$N) temperature in K, b) zonal mean zonal wind ($55°$ to $75°$N) in $ms^{-1}$, c) polar cap LW heating rates in $Kday^{-1}$, and d) polar cap dynamical heating rates in $Kday^{-1}$ with month and height for Chem OFF (contours) and for the differences between Chem ON and Chem OFF (shading). Contour intervals are a) $10$ K, b) $5$ $ms^{-1}$, c) $1$ $Kday^{-1}$, and d) $0.5$ $Kday^{-1}$. Solid contours are used for positive values, dashed contours are used for negative values. The zero contour is omitted. Statistically insignificant areas are hatched at the 95% level.

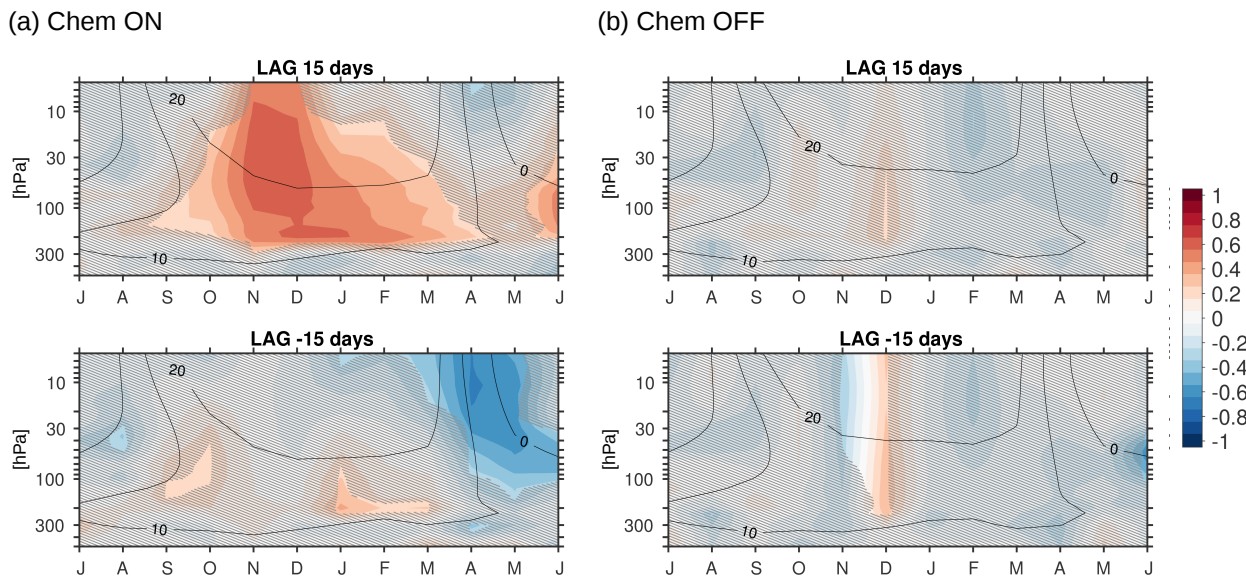

**Figure 5.** Correlation between polar cap (70° to 90° N) ozone at 50 hPa and polar cap dynamical heating rates in a) Chem ON and b) Chem OFF for ozone lagging by 15 days (LAG 15 days) and ozone leading by 15 days (LAG -15 days). Statistically insignificant areas are hatched at the 95% level.

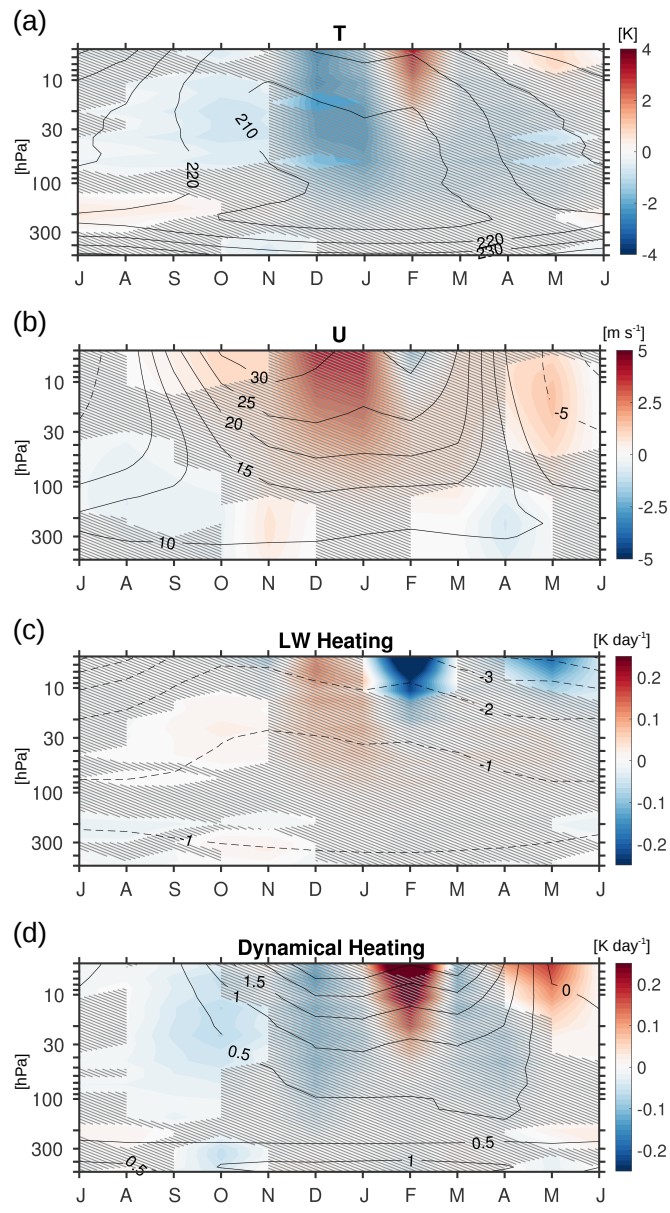

**Figure 6.** Same as Figure 4 but using Chem OFF 3D for comparison to Chem ON.

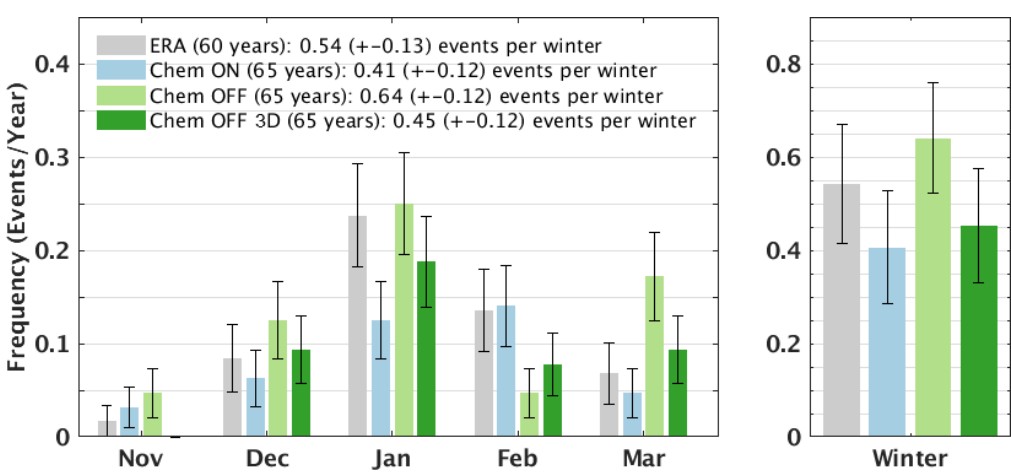

**Figure 7.** Monthly SSW frequency (left) and winter SSW frequency (right) for the combined ERA data (gray), Chem ON (blue), Chem OFF (light green) and Chem OFF 3D (dark green). Error bars are shown in the figure. They indicate the standard error for the monthly frequencies and the 95% confidence interval based on the standard error for the mean winter frequency.

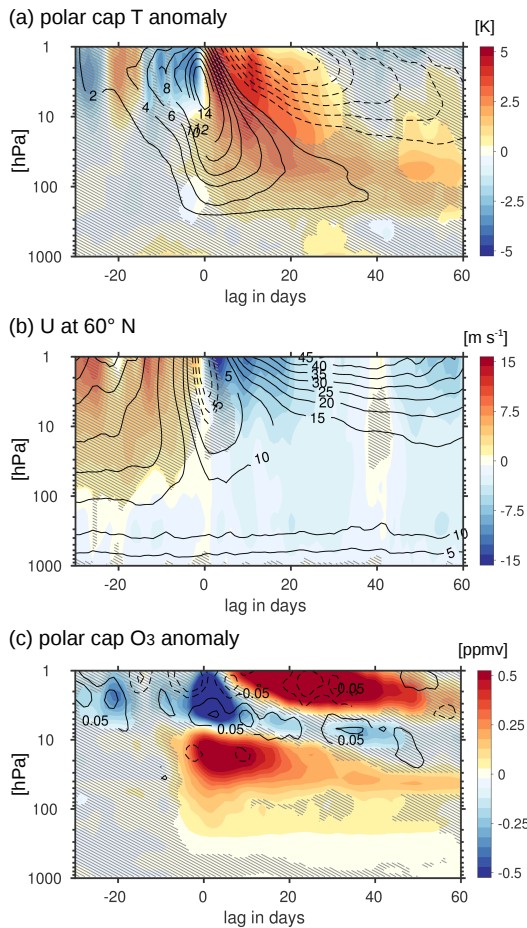

**Figure 8.** SSW composites for a) polar cap (60°to 90°N) temperature anomaly in K, b) zonal mean zonal wind at 60°N in ms$^{-1}$ and for c) polar cap ozone anomaly in ppm with lag in days with respect to the SSW central date (lag 0) and height. Contour lines show the composite for the Chem OFF run. Shading shows the difference between Chem ON and Chem OFF SSW composites. Contour intervals are a) 2 K, b) 5 ms$^{-1}$, and c) 0.05 ppmv. Solid contours are used for positive values, dashed contours are used for negative values. The zero contour is omitted. Statistically insignificant areas are hatched at the 90% level (two-sample t-test).

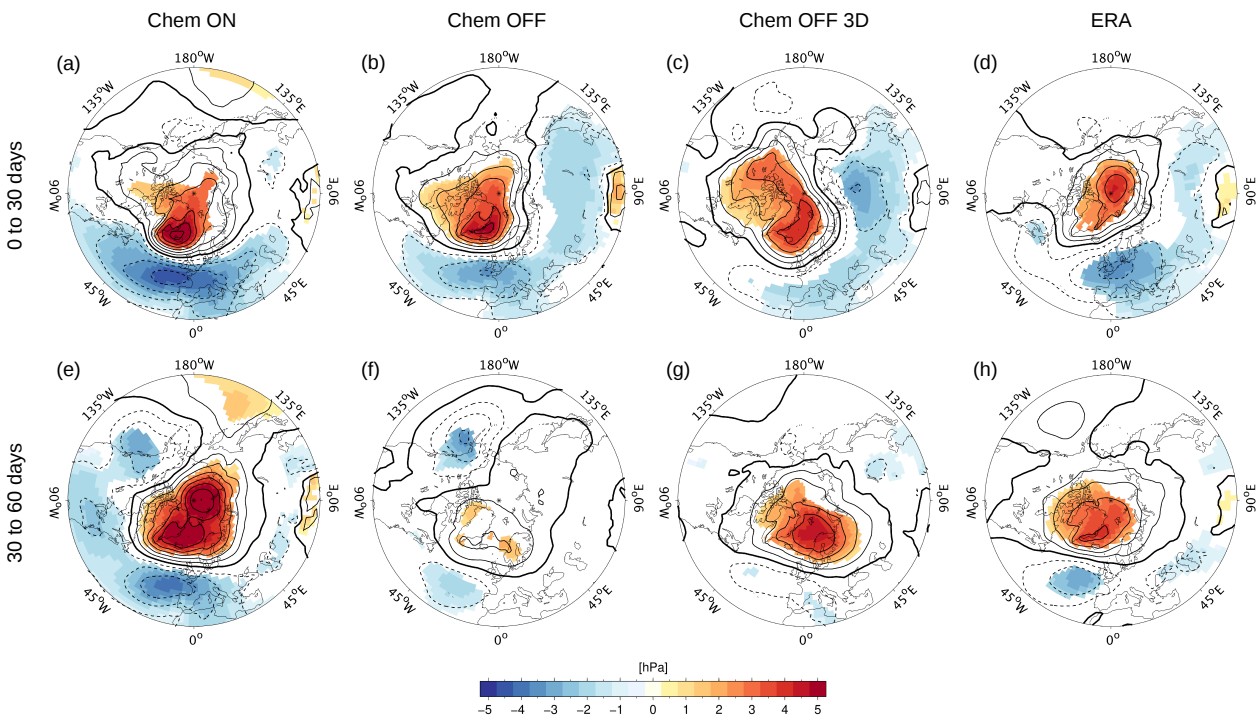

**Figure 9.** SSW composite of SLP anomalies in hPa averaged over 0 to 30 days (a, b, c and d) and over 30 to 60 days (e, f, g and h) following the central date of the SSW for a) and e) Chem ON, b) and f) Chem OFF, c) and g) Chem OFF 3D, and d) and h) combined ERA data. Contour lines show the full composites, while only statistically significant areas at the 95% level are colored. Solid contours are used for positive values, dashed contours are used for negative values. The zero contour is a bold solid line. The contour line interval is 1 hPa.

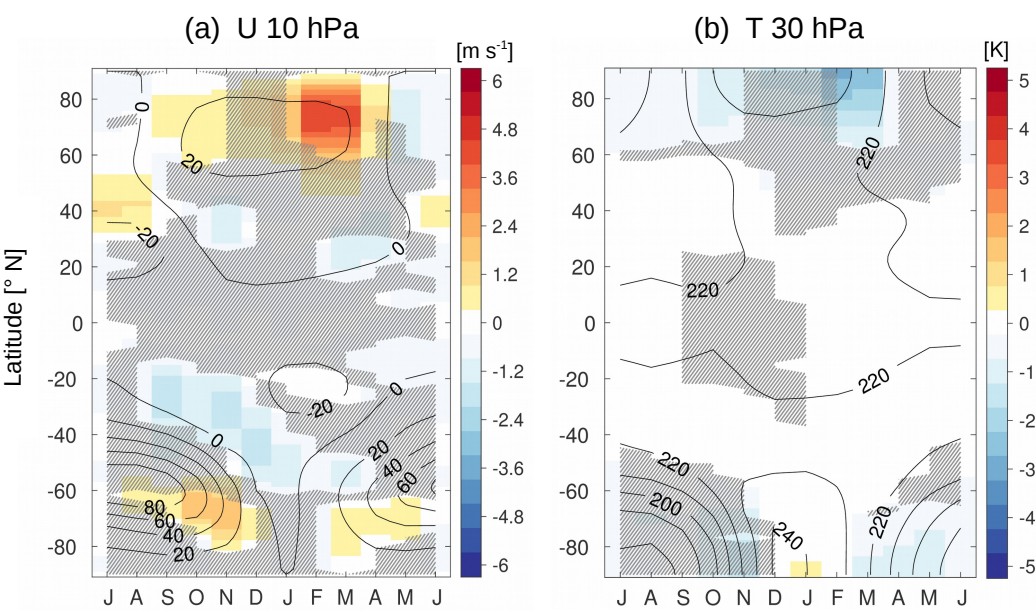

**Figure 10.** Climatological zonal mean a) zonal wind at 10 hPa in ms$^{-1}$, and b) temperature at 30 hPa in K with month and latitude for Chem OFF CTRL (contours) and for the difference between Chem ON CTRL and Chem OFF CTRL (shading). Contour intervals are a) 20 ms$^{-1}$, and b) 10 K. Statistically insignificant areas are hatched at the 95% level.

**Table 1.** Model experiments carried out with CESM1(WACCM) in Chem ON, Chem OFF and Chem OFF 3D mode. For more details see text.

| Experiment/ Data | Ozone setting | Years | SSWs during winters of | |
|---|---|---|---|---|
| | | | 1955/56 to 2018/19 | 1958/59 to 2016/17 |
| Chem ON | interactive | 1955 to 2019 | 26 | 24 |
| Chem OFF | prescribed* zonal mean | 1955 to 2019 | 41 | 40 |
| Chem OFF 3D | prescribed* zonally asymmetric | 1955 to 2019 | 30 | 28 |
| ERA | - | 1958 to 2017 | - | 32 |

* The ozone data used for prescription originates from the Chem ON run.