# Peer review of "The importance of interactive chemistry for stratosphere–troposphere coupling"

_Atmospheric Chemistry and Physics, 2018_

## Referee Comment (RC1) · Anonymous Referee #1 · 19 Nov 2018

The paper is generally well-written and well-organized. In particular, I like how the authors make an attempt at explaining the full dynamical storyline rather than just relying on statistical metrics. However, there are also several points that require improvement. For example, the importance of interactive chemistry has been realized in many other types of simulations and contexts, which is not mentioned at the moment. In addition, certain scientific aspects need revision and/or clarification (see general comments). More specific points and technical corrections (typos etc) are grouped separately. Subject to these revisions, I would recommend publication.

[Figure]

**General comments:**

1. The importance of interactive chemistry has been shown in many contexts other than in relation to the polar vortices. A non-exhaustive list of examples includes effects on global climate sensitivity and the Walker circulation (e.g. Dietmueller et al. 2014, Chiodo & Polvani 2016, Nowack et al. 2017, Noda et al. 2018) as well as the mid-latitude jet-streams (Chiodo & Polvani 2017, Nowack et al. 2018). This wider context should be highlighted briefly either in the introduction or discussion section.

2. A wider perspective would further allow the authors to discuss the relevance of certain climate feedbacks such as changes in stratospheric water vapor. The authors discuss the feedback loops (Figure 1) purely from the perspective that ozone depletion leads to cooling and corresponding changes in the zonal wind and wave propagation, which in turn affects temperatures and ozone due to changes in meridional transport of heat and ozone. However, the same changes would affect the transport of, for example, water vapor into the vortex, which is important for PSC formation (cf. winter 2011) and temperatures in the lower stratosphere (more water vapor, more longwave cooling). Do you find such dynamically-induced changes in water vapor and how would these qualitatively modulate the described feedbacks? Finally, I assume that stratospheric water vapor anomalies due to historic stratosphere-reaching volcanic eruptions are similar in both the interactive and non-interactive simulations, as you take ozone time series from the interactive runs. However, are there any significant differences in the background water vapor levels between the interactive and non-interactive simulations?

3. Concerning volcanic eruptions and the way you prescribed ozone (e.g. mentions on p.1 l.16-18, p.5 l.1-14): while the forcings are the same, the free-running sea surface temperatures from the interactive ocean are not. a) How do you

think this affects the dissimilarities/occurrence of the SSWs between the runs? b) Could the use of the model-consistent ozone time series be a reason why you find that 3D climatologies work fairly well? I assume that taking a 3D ozone field from another model would much more negatively affect the vortex climatology. In that case, you would have to recommend model-consistent 3D ozone forcings, which are much harder to produce (i.e. why not run interactively anyway in that case)? Finally, most models would also not use a daily updated ozone forcing, which could lead to even larger differences than those found here. These details in the set-up need more discussion/context beyond what you have done here...otherwise general climate modelers will just take the next best 3D climatology.

4. The study would greatly benefit from more ensemble members for each run, which could consolidate many of the conclusions reached. From my side, this is to be seen as a recommendation rather than a request. However, this could help overcome some of the significance issues (as raised by the authors themselves in the conclusions and as is clear from Figure 7).

**Specific comments:**

1. p.1 l.23: ...ozone is MAINLY produced in the tropics; cf. Grewe 2006.

2. p.2 l. 2-4: it is not just ozone absorbing but also the production of ozone from molecular oxygen.

3. p.2 l.33: slightly awkward sentence with confused reasoning. Maybe: "The impact of ozone depletion on...spring (when sunlight returns) and, following our above discussion, will be very sensitive to the background state of the polar vortex.

4. next sentence: ...into the summer circulation, thus implying enhanced wave prop-
agation (dynamic heating) as a result of ozone-depletion-cooling (?). Link it back
to the discussion. In the next sentence, I am not sure any more what you exactly
mean by 'negative feedback' (p.3, l.1).

5. p.3 l.8-16: you mention all these different treatments of ozone, but you actually
use a different way of a time-dependent model-consistent ozone. Therefore, you
also explore a different error space than if you had used the climatology by Cionni
et al. This paragraph leads the reader on the wrong track, see also my general
comment 3. This point requires additional clarification here, in the abstract, and
in the conclusions.

6. p.3 l.24 Son et al. 2008 would be another good citation here

7. p.4 l.2-3: there are only a few studies that are designed to systematically compare
the effect of including and excluding interactive chemistry in the same model. See
my general comment 1. This might be true in this context but all studies I mention
there did indeed the same, just focusing on different phenomena.

8. p.4 l.15-21: could the authors say more about why the various studies found
different results. Did they use different climatological ozone fields? Coupled
oceans? Stratospheric resolutions? Is it simply dependent on the chemistry-
climate model used? Equilibrium vs transient runs?

9. p.4 l.33-35: so how specifically is your approach different to the one used else-
where in terms of quantifying surface impacts in particular, or is it just the STC
you are referring to?

10. p.5/6 model description: were all runs initiated from the same ocean spin-up run
in 1950?

11. p.6 l. 22: how is it possible to prescribe the total heating rates? I understand correctly that this applies only above 65km?

12. p.6 l.25: specify all necessary components, is this next to ozone also methane,... Here you mean the entire atmosphere again, not just above 65km?

13. p.7 l.2-4: I find this quite a non-standard procedure for calculating the anomaly. Is this the global mean for that year? Please add some more detail rather than just referring to a reference.

14. p.7 l. 6: slightly awkward formulation. 1hPa is the entire stratosphere.

15. p.7 l.16/17: Why would you omit the criterion if it has no influence. In that case, you might just as well say you included it.

16. p.9 l. 34: Since a statistically[...]. You say that but don't actually show it. However, I would indeed be interested in seeing those correlation plots from CHEM-OFF as well. Can you put them just next to the other plots in Figures 5? That would be quite convincing!

17. p.10 l.10-12: could you iterate a bit more here. Which other processes do you have in mind? Could water vapor play a role? How would ozone waves specifically perturb the picture that you outlined before? Enhancing local dynamical wave propagation?

18. Figure 8: I find the different effects impacting these results difficult to comprehend. As you show in Figure 7, the timing of SSWs occurring in CHEM-ON and CHEM-OFF is very different. Could these average changes simply be due to different background states (many CHEM-OFF events happen later during the year) between these two cases, affecting downward propagation? Could you maybe provide a similar plot just for January and February when you have a similar number of events in total?

19. I would recommend adding results for CHEM-OFF-3D to Figure 7. How is the timing of events in that case?

20. Further recent studies that could be cited: Silverman et al. (2018) when talking about ozone waves and Nowack et al. (2018)b when talking about possible alternative representations of ozone.

**Technical corrections:**

1. p.1 l.9: ...statistically significantly...

2. p.2 l.6: revise: ...over the thermal wind balance...

3. p.2 l.8: ...and, by extension, surface climate...

4. p.3 l.2: swap 'accurate' to 'sophisticated'

5. p.3 l.5: reformulate, for example: However, fully interactive atmospheric chemistry schemes are computationally expensive. (...the ocean is completely separate, so not sure why to mention...). An alternative way...

6. p.3 l.19: ...once sunlight returns...

7. p.5 l.27: ...chemistry-climate...

8. p.5 l.30: On the SH (?), cold bias in the stratosphere, or surface, or where?

9. p.7 l.10: italicize 'A'

10. p.10 l.6: typo

11. p.10 l.22: fewer SSWs
**References:**

1. Dietmüller, S., Ponater, M., Sausen, R. (2014). Interactive ozone induces a negative feedback in CO2-driven climate change simulations. Journal of Geophysical Research: Atmospheres, 119, 1796–1805. http://doi.org/10.1002/2013JD020575.

2. Chiodo, G., Polvani, L. M. (2016). Reduction of climate sensitivity to solar forcing due to stratospheric ozone feedback. Journal of Climate, 29, 4651–4663. http://doi.org/10.1175/JCLI-D-15-0721.1.

3. Nowack, P. J., Braesicke, P., Abraham, N. L., Pyle, J. A. (2017). On the role of ozone feedback in the ENSO amplitude response under global warming. Geophysical Research Letters, 44, 3858–3866. http://doi.org/10.1002/2016GL072418.

4. Noda, S., Kodera, K., Adachi, Y., Deushi, M., Kitoh, A., Mizuta, R., . . . Yoden, S. (2018). Mitigation of global cooling by stratospheric chemistry feedbacks in a simulation of the Last Glacial Maximum. Journal of Geophysical Research: Atmospheres, 123, 9378–9390. http://doi.org/10.1029/2017JD028017.

5. Chiodo, G., Polvani, L. M. (2017). Reduced Southern Hemispheric circulation response to quadrupled CO2 due to stratospheric ozone feedback. Geophysical Research Letters, 43, 1–10. http://doi.org/10.1002/2016GL071011.

6. Nowack, P. J., Abraham, N. L., Braesicke, P., Pyle, J. A. (2018). The impact of stratospheric ozone feedbacks on climate sensitivity estimates. Journal of Geophysical Research: Atmospheres, 123(9), 4630–4641. http://doi.org/10.1002/2017JD027943.

7. Grewe, V. (2006). The origin of ozone. Atmospheric Chemistry and Physics, 6, 1495–1511.

8. Son, S.-W., Polvani, L. M., Waugh, D. W., Akiyoshi, H., Garcia, R., Kinnison, D., . . . Shibata, K. (2008). The impact of stratospheric ozone recovery on the Southern Hemisphere westerly jet. Science, 320, 1486–1489. http://doi.org/10.1126/science.1155939

9. Silverman, V., Harnik, N., Matthes, K., Lubis, S. W., Wahl, S. (2018). Radiative effects of ozone waves on the Northern Hemisphere polar vortex and its modulation by the QBO. Atmospheric Chemistry and Physics, 18, 6637–6659. http://doi.org/10.5194/acp-2017-641. âĄă

10. Nowack, P., Braesicke, P., Haigh, J., Abraham, N., Pyle, J., Voulgarakis, A. (2018). Using machine learning to build temperature-based ozone parameterizations for climate sensitivity simulations. Environmental Research Letters, 13, 104016. http://doi.org/10.1088/1748-9326/aae2be

âĄăâĄăâĄăâĄăâĄăâĄăâĄă
* * *

---

## Referee Comment (RC2) · Anonymous Referee #2 · 22 Nov 2018

Review of manuscript ACP-2018-1052: "The importance of interactive chemistry for stratosphere–troposphere–coupling", by Sabine Hasse and Katja Matthes.

This paper aims to evaluate the effects of including interactive chemistry in the representation of the Northern Hemisphere stratosphere-troposphere coupling in a global model. The methodology consists on analyzing three 65-year runs, each of which is generated using different configurations of a chemistry climate model WACCM: one with full interactive chemistry (the standard model setup, or Chem On), and two with prescribed chemistry (one with zonal-mean ozone fields, or Chem Off, and the other with 3D ozone fields, or Chem Off 3D). The authors find differences in both the clima-

tology and the interannual variability of the stratosphere in these three runs, which they attribute to the different representation of chemistry-dynamics feedbacks in the three model setups. In particular, they find a negative feedback between lower stratosphere ozone concentrations and stratospheric dynamics in late winter and spring in Chem On, which may explain differences in mean wind and temperature in this season in the three model setups. They also find that the temperature anomalies associated with sudden stratospheric warmings (SSWs) in the lower stratosphere last longer in Chem On than in Chem Off, which it is attributed to the the radiative effects of interactive ozone in Chem On.

This is a nice attempt to address this complex problem of evaluating the effects of having interactive chemistry in the interannual variability of the stratosphere in the NH. The manuscript is well written, and results put into context. I detail below some comments that the authors may want to address before I recommend the paper for publication.

General comments:

1) I have a general concern about comparing different configurations of WACCM and attributing differences in the variability (for example, the intraseasonal distribution of SSWs) to the different model configurations (interactive versus fixed chemistry). The three model setups have somewhat different basic states of wind and temperature, which may condition the stratospheric variability in each setup. The authors should recognize (if I understand correctly) that the standard configuration of WACCM (Chem On) has been exhaustively tuned by NCAR modelers in order to produce the best possible climatology. When "downgrading" the model by specifying the evolution of ozone (and other species) in Chem Off, the resulting climatology may not be optimized (in the stratosphere, I am mainly talking about tuning the gravity wave drag parameterizations). The fact that we see a wintertime NH stratosphere that is systematically colder in Chem On may be (at least in part) a consequence of not having an optimized setup in Chem Off / Chem Off 3D.

But I wonder whether part of the U and T differences shown in Figs. 2 and 4 come from the fact that there are large differences in the number of SSWs among the different model setups (Chem Off has nearly twice as many SSWs as Chem On does). It would be interesting to produce new Figs. 2 and 4, but selecting years without SSWs. This will give us more comparable basic states. And it would be interesting to compare the amplitude of the waves as well –the vertical component of the EP flux at 300 hPa and 100 hPa, or v'T', are widely-used options. If the wave characteristics in these undisturbed-vortex years are similar among the different model setups, and the zonal-mean differences in U and T are reduced, particularly in winter when we should not expect large differences in the radiative forcing between Chem On and Off, then I would be convinced that the U and T differences already shown in those figures reflect the effects of interactive chemistry in the model.

2) I miss in this study further comparisons with reanalysis data, not only in the frequency of SSWs. I assume that the standard version of WACCM (Chem On) is the one that better compares with reanalysis, but it would be interesting and clarifying to add panels to Figs. 2, 3, 4, 8 and 9 comparing Chem On/Off with ERA fields.

3) More generally, I wonder whether one realization of each configuration of the model (Chem On / Chem Off / Chem Off 3D) is enough to draw robust conclusions on the role of chemistry-dynamics interactions in the highly variable NH stratosphere. I understand that producing new runs to perform an ensemble analysis is time-demanding, so I leave it to the authors' discretion whether attack this issue or not. In any case, a discussion about the limitations of working with only one realization would add quality to the article.

Minor comments:

- Page 3, line 15. I do not think there is enough information in Fig. 2 as to discriminate between the deep and the shallow branch.

- Page 3, line 31. I guess "dynamical heating" refers to temperature advection by the mean residual circulation, but please define it (perhaps in the methods section?).

[Figure]

- Figure 5. This is a nice figure that helps elucidate cause and effect in the interaction between temperature and ozone in Chem On. I have one question. If ozone concentrations lead the dynamical heating in spring, as suggested in Fig. 5b, should we not expect those negative correlations to show up in Chem Off and Chem Off 3D (since ozone from Chem On is prescribed)? Is it the case?

- Page 9, line 4-5/33-34. Please explain why you say that the final warming is more intense in Chem On. I would expect earlier final warmings to be "more dynamical", in the sense that the radiative forcing is still weak in late winter or early spring, and hence more abrupt and intense. And the opposite for late final warmings.

Technical comments:

- Title: stratosphere-troposphere coupling?

- Page 9, line 21. "as well as".

---

## Author Response (AR1)

Dear Farahnaz Khosrawi, dear reviewers,

We would like to thank you for your very valuable suggestions and comments. We give detailed answers to the specific points of the two reviewers highlighted in blue below.
At the end of this document, a revised version of the manuscript is attached. All changes made in this manuscript are highlighted in yellow[1].

Best regards,
Sabine Haase and Katja Matthes
* * *
[1] This does unfortunately not apply for references. We hope you excuse this inconvenience.

**Anonymous Referee #1**

The paper is generally well-written and well-organized. In particular, I like how the authors make an attempt at explaining the full dynamical storyline rather than just relying on statistical metrics. However, there are also several points that require improvement. For example, the importance of interactive chemistry has been realized in many other types of simulations and contexts, which is not mentioned at the moment. In addition, certain scientific aspects need revision and/or clarification (see general comments). More specific points and technical corrections (typos etc) are grouped separately. Subject to these revisions, I would recommend publication.

Thank you very much for your constructive remarks. We address your issues in detail below.

**General comments:**

1. The importance of interactive chemistry has been shown in many contexts other than in relation to the polar vortices. A non-exhaustive list of examples includes effects on global climate sensitivity and the Walker circulation (e.g. Dietmueller et al. 2014, Chiodo & Polvani 2016, Nowack et al. 2017, Noda et al. 2018) as well as the mid-latitude jet-streams (Chiodo & Polvani 2017, Nowack et al. 2018). This wider context should be highlighted briefly either in the introduction or discussion section.

Thank you for pointing out these additional references. We extended the discussion about interactive chemistry in the introduction of the paper, focusing on the references you recommended.

The following section is now included:

"Since the interactive chemistry module in a climate model is computationally very expensive, it is necessary to elucidate alternative representations of in particular ozone for long–term climate simulations. So far, the importance of interactive chemistry in climate models has been evaluated mainly for experimental settings that focused on the effect of an altered external forcing, such as a change in solar irradiance or in $CO_2$ concentrations (e.g., Chiodo and Polvani, 2016, 2017; Dietmüller et al., 2014; Noda et al., 2018; Nowack et al., 2017, 2018). In these studies CCM simulations were compared to model simulations forced with a constant ozone field (e.g. based on pre-industrial control conditions), which did not include the ozone response to the changing external forcing. It was shown that the ozone response to the external forcing has an important damping effect onto the surface climate response to the external forcing. Namely, under such conditions, including interactive chemistry reduces the model's climate sensitivity (e.g., Chiodo and Polvani, 2016; Dietmüller et al., 2014; Noda et al., 2018; Nowack et al., 2018) and connected surface responses, such as the tropospheric jet (e.g., Chiodo and Polvani, 2017) or ENSO trends (e.g., Nowack et al., 2017)."

2. A wider perspective would further allow the authors to discuss the relevance of certain climate feedbacks such as changes in stratospheric water vapor. The authors discuss the feedback loops (Figure 1) purely from the perspective that ozone depletion leads to cooling and corresponding changes in the zonal wind and wave propagation, which in turn affects temperatures and ozone due to changes in meridional transport of heat and ozone. However, the same changes would affect the transport of, for example, water vapor into the vortex, which is important for PSC formation (cf. winter 2011) and temperatures in the lower stratosphere (more water vapor, more longwave cooling). Do you find such dynamically-induced changes in water vapor and how would these qualitatively modulate the described feedbacks? Finally, I assume that stratospheric water vapor anomalies due to historic stratosphere-reaching volcanic eruptions are similar in both the interactive and non-interactive simulations, as you take ozone time series from the interactive

runs. However, are there any significant differences in the background water vapor levels between the interactive and non-interactive simulations?

Of course, we agree that other trace gases, such as water vapor, are affected by transport and mixing processes as well. Water vapor is interactively represented only in the Chem ON simulation and so is a potential effect on PSCs. In the Chem OFF simulation for each methane molecule lost, two water molecules are generated. $CH_4$ is prognostic in Chem OFF, specified at the surface based on Meinshausen et al. (2011), transported by the calculated wind field and removed from the model using loss rates taken from Garcia and Solomon (1994) as described in Smith et al. (2014).

Due to the different representation of water vapor in the simulations, we cannot apply the same analysis for water vapor as we did for ozone. The impact that PSC formation plays for ozone depletion is included in the prescribed ozone field (and so are impacts by volcanic aerosols etc.), whereas the long-wave effect of water vapor would differ between the simulations based on the differences in water vapor concentrations. For your reference, we depicted the characteristics of middle atmosphere water vapor using the tape recorder representation in Figure A 1. The tape recorder shows that water vapor agrees well between Chem ON and Chem OFF. Differences are low and are shown for the Northern Hemisphere in Figure A 2 together with the long-wave heating rate differences that are also shown in the manuscript. It becomes obvious that there are significant differences in water vapor (Fig. A 2a), but these differences are low in amplitude and do not seem to affect the differences in long-wave heating (Fig. A 2b).

In the manuscript we now mention a possible contribution of water vapor. However, since ozone has a much higher potential to influence stratospheric temperatures and therefore also model dynamics we keep the focus of our discussion on ozone in this publication.

Introduction:
"Although other trace gases, such as water vapor, can also be affected by these feedbacks, we concentrate our discussion on ozone in this publication. The effects of ozone can be represented differently in climate models:"

[Figure]

**Figure A 1:** *Water vapor tape recorder signal in Chem ON and Chem OFF. Plots show the water vapor anomaly in [ppmv] from the time-mean averaged over 10°S to 10°N.*

[Figure]

**Figure A 2:** *Climatological Difference Chem ONn minus Chem OFF for a) water vapor in [%] and b) LW heating in [K/day]. The contours depict the climatology of the Chem OFF run. Non-hatched areas show significance at the 95% level.*

3. Concerning volcanic eruptions and the way you prescribed ozone (e.g. mentions on p.1 l.16-18, p.5 l.1-14): while the forcings are the same, the free-running sea surface temperatures from the interactive ocean are not. a) How do you think this affects the dissimilarities/occurrence of the SSWs between the runs? b) Could the use of the model-consistent ozone time series be a reason why you find that 3D climatologies work fairly well? I assume that taking a 3D ozone field from another model would much more negatively affect the vortex climatology. In that case, you would have to recommend model-consistent 3D ozone forcings, which are much harder to produce (i.e. why not run interactively anyway in that case)? Finally, most models would also not use a daily updated ozone forcing, which could lead to even larger differences than those found here. These details in the set-up need more discussion/context beyond what you have done here...otherwise general climate modelers will just take the next best 3D climatology.

a) The interactive ocean can of course have an effect onto stratospheric dynamics. Since, the El Nino Southern Oscillation (ENSO) is regarded as one of the most important variability patterns that do influence stratospheric dynamics we here show the ENSO index for the different simulations in Figure A 3. ENSO variability is very similar between the simulations. We therefore do not expect a large impact onto the differences that we find between Chem ON and Chem OFF by the interactive ocean but cannot rule out that differences during mid winter might be influenced by the ocean.

b) Yes, using model consistent ozone forcing might be crucial for our results. We now give more emphasis to this fact and discuss this aspect in the manuscript (please, see our answer to your specific comment #5 for changes in the manuscript). We adapted our recommendation and also highlight the importance of the daily representation of ozone.

**Nino 3.4 Index**

[Figure]

**Figure A 3:** *Nino 3.4 index for a) Chem ON, b) Chem OFF and c) Chem OFF 3D. The standard deviation for the whole period and for the period of 1960 to 1990 is given in each subplot.*

4. The study would greatly benefit from more ensemble members for each run, which could consolidate many of the conclusions reached. From my side, this is to be seen as a recommendation rather than a request. However, this could help overcome some of the significance issues (as raised by the authors themselves in the conclusions and as is clear from Figure 7).

To further validate our results we now additionally include the results from a longer control model simulation with constant greenhouse gases and ozone depleting substances in the discussion part of the paper. We agree that a larger number of simulations would be very favorable and are planning on a larger ensemble for a follow-up study.

**Specific comments:**

1. p.1 l.23: ...ozone is MAINLY produced in the tropics; cf. Grewe 2006.

   it now reads:
   "… ozone is mainly produced in the tropics"

2. p.2 l. 2-4: it is not just ozone absorbing but also the production of ozone from molecular oxygen.

   Yes, we include the chemical heating now as well:
   "The production of ozone and the absorption of UV radiation by stratospheric ozone leads to the characteristic increase of stratospheric temperature with height resulting in a stable stratification."

3. p.2 l.33: slightly awkward sentence with confused reasoning. Maybe: "The impact of ozone depletion on...spring (when sunlight returns) and, following our above discussion, will be very sensitive to the background state of the polar vortex.

Thank you for the remark. We adapted the sentence accordingly.

"The impact of ozone depletion on stratospheric dynamics is strongest during spring (when solar irradiance is available to initiate ozone depletion) and, following our above discussion, will be very sensitive to the background state of the polar vortex. In fact, previous studies suggested a dominance of the negative feedback during the vortex break down […]"

4. next sentence: ...into the summer circulation, thus implying enhanced wave propagation (dynamic heating) as a result of ozone-depletion-cooling (?). Link it back to the discussion. In the next sentence, I am not sure any more what you exactly mean by 'negative feedback' (p.3, l.1).

See previous comment.

5. p.3 l.8-16: you mention all these different treatments of ozone, but you actually use a different way of a time-dependent model-consistent ozone. Therefore, you also explore a different error space than if you had used the climatology by Cionni et al. This paragraph leads the reader on the wrong track, see also my general comment 3. This point requires additional clarification here, in the abstract, and in the conclusions.

As mentioned in the answer to your general comment 3, we extended the discussion about the ozone forcing used and gave mire emphasis to the fact that we use a model-consistent ozone forcing in the zonal mean as well as in the zonally asymmetric forcing.

In the manuscript, we added or adapted the following paragraphs:

Here:

"When prescribing ozone as monthly mean, zonal mean fields, some aspects of ozone variability, such as zonal asymmetries in ozone, are neglected. Using a monthly climatology was shown to introduce biases in the model's ozone field that reduce the strength of the actual seasonal ozone cycle due to the interpolation of the prescribed ozone field to the model time step (Neely et al., 2014). To avoid these biases, a daily ozone forcing can be applied. Furthermore, ozone is not distributed zonally symmetric in the real atmosphere, therefore prescribing zonal mean ozone values inhibits the effect that zonal asymmetries in ozone, also referred to as ozone waves, can have onto the dynamics. Different studies showed that including zonal asymmetries in ozone in a model simulation would lead to a warmer and weaker stratospheric polar vortex in the NH, which was also associated with a higher frequency in SSWs (e.g., Gabriel et al., 2007; Gillett et al., 2009; McCormack et al., 2011; Peters et al., 2015). The recommended ozone forcing for CMIP6 now includes zonal asymmetries, but does not include variability on time scales smaller than a month (Checa-Garcia et al., 2018).

Since the interactive chemistry module in a climate model is computationally very expensive, it is necessary to elucidate alternative representations of in particular ozone for long–term climate simulations. So far, the importance of interactive chemistry in climate models has been evaluated mainly for experimental settings that focused on the effect of an altered external forcing, such as a change in solar irradiance or in $CO_2$ concentrations (e.g., Chiodo and Polvani, 2016, 2017; Dietmüller et al., 2014; Noda et al., 2018; Nowack et al., 2017, 2018). […] Here, we use a different approach. We are interested in how feedbacks between ozone chemistry and model dynamics can impact the stratospheric mean state and variability given that the variability in stratospheric ozone is the same between the interactive and specified chemistry experiments. This question will be addressed in the present study by using a time--evolving, model--consistent ozone forcing in the specified chemistry version of the model."

In the abstract:

"To be able to focus on differences that arise from two-way interactions between chemistry and dynamics in the model, the specified chemistry model version uses a time-evolving, model-consistent ozone field generated by the interactive chemistry model version. […]

The results from the zonally asymmetric ozone simulation are closer to the interactive chemistry simulations, implying that under a model-consistent ozone forcing, a three-dimensional representation of the prescribed ozone field is desirable. This suggests that a 3D ozone forcing as recommended for the upcoming CMIP6 simulations has the potential to improve the representation of stratospheric dynamics and chemistry."

In the conclusions:

"Therefore, an interactive chemistry climate model was compared to the specified chemistry version of the same model using a time-evolving, model-consistent, daily ozone forcing. […]

It is however essential to better understand the role of chemistry-dynamics-interactions in order to improve our decisions about how ozone shall be prescribed in upcoming model simulations. A new approach was recently presented by Nowack et al. (2018a), who discuss the potential of machine learning to parameterize the impact of ozone in different standard scenarios, such as in a $4xCO_2$ setting. Based on our findings from prescribing a model-consistent, daily ozone forcing, we argue that a 3D ozone forcing as now provided for CMIP6 has the potential to improve the representation of the impact that ozone chemistry has on model dynamics. However, such a forcing does not perfectly compare to our experimental setting since the more generalized CMIP6 ozone forcing cannot supply model-consistent ozone fields for different models and is based on monthly mean data."

6. p.3 l.24 Son et al. 2008 would be another good citation here

   We agree and include Son et al. 2008 as an additional reference at this point.

7. p.4 l.2-3: there are only a few studies that are designed to systematically compare the effect of including and excluding interactive chemistry in the same model. See my general comment 1. This might be true in this context but all studies I mention there did indeed the same, just focusing on different phenomena.

   We adapted the sentence to better clarify what we mean. It now reads:
   "There are only a few studies, like that of Li et al. (2016), that are designed to systematically compare the effect of including or excluding interactive chemistry in the same model, i.e. using the ozone forcing from the CCM also in the specified chemistry version of the model. "

   We include the studies mentioned in your general comment above and discuss this in more detail now. Please, see our answer to your general comment #1.

8. p.4 l.15-21: could the authors say more about why the various studies found different results. Did they use different climatological ozone fields? Coupled oceans? Stratospheric resolutions? Is it simply dependent on the chemistry climate model used? Equilibrium vs transient runs?

   The paper by Natalia Calvo and colleagues was the only one using an interactive chemistry model. All the other studies mentioned here prescribed ozone. Calvo et al. 2015 argue that including interactive chemistry in their study would be the reason for the significant surface impact of low ozone years on the NH under historical conditions. Apart from not including interactive chemistry, the studies by Cheung et al. (2014), Karpechko et al. (2014) and Smith and Polvani (2014) were set up differently also in other aspects:

1. Cheung et al. (2014) examined the potential benefits from using zonal mean ozone data from the Earth Observing System (EOS) Microwave Limb Sounder (MLS) for medium-extended range tropospheric forecasts. Specifically for the winter 2011, they found a general reduction in forecast error in the stratosphere with this improved ozone forcing. In the troposphere though, they did not find a significant reduction in the root-mean-square forecast errors.

2. Karpechko et al. (2014) used ECHAM5 simulations to investigate the surface response of the ozone anomalies from September 2010 to April 2011 superimposed onto a climatological ozone forcing used in a control simulation. The prescribed monthly zonal mean ozone anomaly was calculated from MERRA data. This experiment was compared to an experiment using prescribed SST anomalies instead of ozone and to an experiment using both forcings. The authors found that significant surface responses only occur when SST and ozone anomalies are prescribed together in their model setup.

3. Smith and Polvani (2014) contrasted model simulations (using CAM3) with positive and negative springtime ozone anomalies. The prescribed ozone fields are based on the IGAC/SPARC ozone database, to which synthetic ozone anomalies have been added. These anomalies are based on MERRA data. For ozone anomaly amplitudes somewhat larger than the recent observed variability, they find a significant influence on the tropospheric circulation (surface temperatures and precipitation patterns). They did find a significant surface signal for anomalies within the observed range.

9. p.4 l.33-35: so how specifically is your approach different to the one used elsewhere in terms of quantifying surface impacts in particular, or is it just the STC you are referring to?

Yes, we specifically refer to the STC at this point. In general the different experimental design, though, that is now discussed in more detail above is something new in itself.

10. p.5/6 model description: were all runs initiated from the same ocean spin-up run in 1950?

Yes, all runs were initiated from the same initial conditions in the atmosphere and ocean on January 1$^{st}$ in 1955.

11. p.6 l. 22: how is it possible to prescribe the total heating rates? I understand correctly that this applies only above 65km?

Yes, it applies only above approximately 65 km. Heating rates are as well as species concentrations taken from the Chem ON output. The prescribed total short-wave and chemical heating rates also incorporate heating from the directly thermalized energy realized during photolysis, photoabsorption and photoionization, and during energetic particle precipitation in the aurora (Smith et al. 2014). This is described in more detail in Smith et al. (2014).

12. p.6 l.25: specify all necessary components, is this next to ozone also methane,... Here you mean the entire atmosphere again, not just above 65km?

Only ozone is prescribed in the whole atmosphere. All other constituents, which include O, $O_2$, $CO_2$, H, NO, and total short-wave and chemical heating rates are prescribed only above approximately 65 km.

For the representation of methane see our answer to you general question 2.

In the manuscript, it now reads:
"[…] we use the output from our transient WACCM integration to specify all necessary components in SC-WACCM (i.e. O, $O_2$, $O_3$, NO, H, $CO_2$ and total short-wave and chemical heating rates)."

13. p.7 l.2-4: I find this quite a non-standard procedure for calculating the anomaly. Is this the global mean for that year? Please add some more detail rather than just referring to a reference.

Yes, it is the global mean for that year. This procedure was chosen to be able to concentrate on internal variability rather than climate change trends.
We extended the description about the method in the manuscript as follows:
"When variability is considered we use deseasonalized daily or monthly data by removing a slowly varying climatology after removing the global mean from each grid point each year. This follows the procedure described in Gerber et al. (2010) and is used to omit the effect that may arise from variability on timescales larger than 30 years, such as the signature of global warming. The slowly varying climatology is produced as follows: First, a 60 day low pass filter is applied. Then, for each time step and grid point, a 30 year low pass filter is applied to the smoothed time series. Gerber et al. (2010) describe this procedure in detail and apply it exemplarily."

14. p.7 l. 6: slightly awkward formulation. 1hPa is the entire stratosphere.

Sorry, the number was not correct. We actually confine the results shown to the region below 5 hPa. We corrected that in the manuscript.

15. p.7 l.16/17: Why would you omit the criterion if it has no influence. In that case, you might just as well say you included it.

Yes, that is true. We include the criterion now. It makes a small difference in the ERA data.

16. p.9 l. 34: Since a statistically[...]. You say that but don't actually show it. However, I would indeed be interested in seeing those correlation plots from CHEM-OFF as

well. Can you put them just next to the other plots in Figures 5? That would be quite convincing!

Yes, these plots are now included in Figure 5 in the manuscript.

17. p.10 l.10-12: could you iterate a bit more here. Which other processes do you have in mind? Could water vapor play a role? How would ozone waves specifically perturb the picture that you outlined before? Enhancing local dynamical wave propagation?

    We added the following sentences here:

    "Including zonal asymmetries in ozone does allow for stronger anomalies in ozone in general since no averaging is applied and for anomalies that do not center over the pole but affect lower latitudes as well. Hence, ozone waves can influence wave propagation and dissipation pathways possibly leading to a better representation of the effect that ozone has on wave--mean flow interactions in our model setup."

18. Figure 8: I find the different effects impacting these results difficult to comprehend. As you show in Figure 7, the timing of SSWs occurring in CHEM-ON and CHEM-OFF is very different. Could these average changes simply be due to different background states (many CHEM-OFF events happen later during the year) between these two cases, affecting downward propagation? Could you maybe provide a similar plot just for January and February when you
    have a similar number of events in total?

    Below you find a figure excluding the March SSWs (Fig. A 4). It shows basically the same difference between the simulations as shown in Figure 8 of the manuscript. We therefore regard the results to be robust. The differing climatological state can have an impact onto the occurrence of SSWs but also SSWs do have an effect onto the climatological mean state.

[Figure]

**Figure A 4:** *Same as Figure 8 in the manuscript but for a sub selection of SSWs: November to February SSWs (left) and January to February SSWs (right).*

19. I would recommend adding results for CHEM-OFF-3D to Figure 7. How is the timing of events in that case?

We include the results from Chem OFF 3D now in this figure. The difference between Chem ON and Chem OFF 3D is much smaller as compared to Chem OFF.

20. Further recent studies that could be cited: Silverman et al. (2018) when talking about ozone waves and Nowack et al. (2018)b when talking about possible alternative representations of ozone.

Yes, these studies are now mentioned in the manuscript as well. Silverman et al. is referred to in the introduction, while Nowack et al. is used in the discussion.

**Technical corrections:**

1. p.1 l.9: ...statistically significantly...
   corrected

2. p.2 l.6: revise: ...over the thermal wind balance...
   it now reads: "[…] through the thermal wind balance."

3. p.2 l.8: ...and, by extension, surface climate...
   adapted accordingly

4. p.3 l.2: swap 'accurate' to 'sophisticated'
   adapted accordingly

5. p.3 l.5: reformulate, for example: However, fully interactive atmospheric chemistry schemes are computationally expensive. (...the ocean is completely separate, so not sure why to mention...). An alternative way...
   We mention the ocean here since a coupled ocean is often used in climate simulations.
   We reformulated the paragraph in response to your comment above as follows: "However, fully interactive atmospheric chemistry schemes are computationally expensive in particular if also an interactive ocean is used for long--term climate model simulations. An alternative way of representing the effects of ozone

chemistry in a climate model is therefore to prescribe ozone fields which can be based on either observed or modeled ozone concentrations."

6. p.3 l.19: ...once sunlight returns...
   adapted accordingly

7. p.5 l.27: ...chemistry-climate...
   adapted accordingly

8. p.5 l.30: On the SH (?), cold bias in the stratosphere, or surface, or where?
   Now reads: "On the SH, CESM1(WACCM) has a strong cold pole bias in the middle atmosphere, […]"

9. p.7 l.10: italicize 'A'
   corrected

10. p.10 l.6: typo
    corrected

11. p.10 l.22: fewer SSWs
    adapted accordingly

**Anonymous Referee #2**

Review of manuscript ACP-2018-1052: "The importance of interactive chemistry for stratosphere–troposphere–coupling", by Sabine Hasse and Katja Matthes. This paper aims to evaluate the effects of including interactive chemistry in the representation of the Northern Hemisphere stratosphere-troposphere coupling in a global model. The methodology consists on analyzing three 65-year runs, each of which is generated using different configurations of a chemistry climate model WACCM: one with full interactive chemistry (the standard model setup, or Chem On), and two with prescribed chemistry (one with zonal-mean ozone fields, or Chem Off, and the other with 3D ozone fields, or Chem Off 3D). The authors find differences in both the climatology and the interannual variability of the stratosphere in these three runs, which they attribute to the different representation of chemistry-dynamics feedbacks in the three model setups. In particular, they find a negative feedback between lower stratosphere ozone concentrations and stratospheric dynamics in late winter and spring in Chem On, which may explain differences in mean wind and temperature in this season in the three model setups. They also find that the temperature anomalies associated with sudden stratospheric warmings (SSWs) in the lower stratosphere last longer in Chem On than in Chem Off, which it is attributed to the the radiative effects of interactive ozone in Chem On.

This is a nice attempt to address this complex problem of evaluating the effects of having interactive chemistry in the interannual variability of the stratosphere in the NH. The manuscript is well written, and results put into context. I detail below some comments that the authors may want to address before I recommend the paper for publication.

Thank you very much for your constructive remarks. We address your issues in detail below.

**General comments:**

1) I have a general concern about comparing different configurations of WACCM and attributing differences in the variability (for example, the intraseasonal distribution of SSWs) to the different model configurations (interactive versus fixed chemistry). The three model setups have somewhat different basic states of wind and temperature, which may condition the stratospheric variability in each setup. The authors should recognize (if I understand correctly) that the standard configuration of WACCM (Chem

On) has been exhaustively tuned by NCAR modelers in order to produce the best possible climatology. When "downgrading" the model by specifying the evolution of ozone (and other species) in Chem Off, the resulting climatology may not be optimized (in the stratosphere, I am mainly talking about tuning the gravity wave drag parameterizations). The fact that we see a wintertime NH stratosphere that is systematically colder in Chem On may be (at least in part) a consequence of not having an optimized setup in Chem Off / Chem Off 3D.

But I wonder whether part of the U and T differences shown in Figs. 2 and 4 come from the fact that there are large differences in the number of SSWs among the different model setups (Chem Off has nearly twice as many SSWs as Chem On does). It would

be interesting to produce new Figs. 2 and 4, but selecting years without SSWs. This will give us more comparable basic states. And it would be interesting to compare the amplitude of the waves as well –the vertical component of the EP flux at 300 hPa and 100 hPa, or v'T', are widely-used options. If the wave characteristics in these undisturbed-vortex years are similar among the different model setups, and the zonalmean differences in U and T are reduced, particularly in winter when we should not expect large differences in the radiative forcing between Chem On and Off, then I

would be convinced that the U and T differences already shown in those figures reflect the effects of interactive chemistry in the model.

The version of Chem OFF used here was validated by Smith et al. (2014) to well represent the characteristics of Chem ON. We did therefore not expect large differences in the response of the two simulations due to the gravity wave parameterization, especially not

in the region considered in our publication. Gravity wave forcing becomes more important at higher altitudes. Figure B 1 shows the difference in temperature tendency due to gravity wave drag between Chem ON and Chem OFF. Compared to the differences found in long-wave heating rates and in dynamical heating rates, the difference in temperature tendency due to gravity waves is very low, to be more precise more than one order of magnitude lower.

[Figure]

**Figure B 1**: *Climatological Difference Chem ON minus Chem OFF for a) temperature tendency due to gravity waves in [K/day] and b) LW heating rates in [K/day]. The contours depict the climatology of the Chem OFF run. Non-hatched areas show significance at the 95% level.*

Figure B 2 shows the heat flux at 100 hPa for zonal wave numbers one to three averaged over the latitudes 45°N to 75°N. It is obvious that planetary wave forcing is comparable between the simulations.

[Figure]

***Figure B 2:*** *Heat flux (v'T') at 100 hPa averaged over 45°N to 75°N for Chem ON (top), Chem OFF (middle) and Chem OFF 3D (bottom). Shown are the contributions of planetary waves, k= 1, 2 and 3.*

Figure B 3 shows the climatological differences without SSWs in comparison to those with SSWs (from the manuscript). The difference between the simulations is lower but still resembles the basic characteristics that we describe in the manuscript.

[Figure]

***Figure B 3:*** *Climatological difference between Chem ON and Chem OFF for a) zonal mean wind at 10 hPa and b) temperature at 30 hPa (shading). The contours depict the climatological mean for the Chem OFF simulation. Non-hatched areas are significant at the 95 % level. The upper row shows the plots that are also used in the manuscript, whereas the lower row uses only years that are not influenced by SSWs.*

With the additional analysis shown above, we hopefully convinced you that the differences shown in the manuscript reflect the effects of interactive chemistry in the model and are not due to insufficient gravity wave parameterizations.

We include the information about the temperature tendency due to gravity waves in the manuscript now:
"By construction, there are no significant differences in the short--wave (SW) heating rates between Chem ON and Chem OFF that could explain the different temperatures between the models in this region, neither can differences in temperature tendencies due to gravity waves (not shown)."

2) I miss in this study further comparisons with reanalysis data, not only in the frequency of SSWs. I assume that the standard version of WACCM (Chem On) is the one that better compares with reanalysis, but it would be interesting and clarifying to add panels to Figs. 2, 3, 4, 8 and 9 comparing Chem On/Off with ERA fields.

The aim of our study is to compare a model simulation with interactive chemistry to a simulation using prescribed chemistry. We did not focus on the general representation of stratospheric conditions in WACCM. WACCM was shown to well represent stratospheric dynamics in earlier studies. For the validation of WACCM and a detailed comparison to observations or reanalysis data we kindly refer to Marsh et al. (2013). For a validation of SC-WACCM (our Chem OFF simulation) we kindly refer to Smith et al. (2014).
We now additionally include a comparison to ERA data for Figure 9 of the manuscript but do not extent the comparison to observations for the other suggested figures since this was not the focus of our study.

3) More generally, I wonder whether one realization of each configuration of the model (Chem On / Chem Off / Chem Off 3D) is enough to draw robust conclusions on the role of chemistry-dynamics interactions in the highly variable NH stratosphere. I understand that producing new runs to perform an ensemble analysis is time-demanding, so I leave

it to the authors' discretion whether attack this issue or not. In any case, a discussion about the limitations of working with only one realization would add quality to the article.

We do now discuss this issue in more detail in the discussion part of the manuscript and include a new figure based on a longer control simulation with constant greenhouse gases and ozone depleting substances to further validate our results. We agree that a larger number of simulations would be very favorable and are planning on a larger ensemble for a follow-up study.

**Minor comments:**

- Page 3, line 15. I do not think there is enough information in Fig. 2 as to discriminate between the deep and the shallow branch.

Yes, we agree. We say now: "at least the shallow branch" in the manuscript.

- Page 3, line 31. I guess "dynamical heating" refers to temperature advection by the mean residual circulation, but please define it (perhaps in the methods section?).

We include a definition of the dynamical heating rates now. It reads:
"The dynamical heating rates, which describe the total adiabatic heating rates in the model dominated by advection through the vertical component of the residual circulation, w* […]"

- Figure 5. This is a nice figure that helps elucidate cause and effect in the interaction between temperature and ozone in Chem On. I have one question. If ozone concentrations lead the dynamical heating in spring, as suggested in Fig. 5b, should we not expect those negative correlations to show up in Chem Off and Chem Off 3D (since ozone from Chem On is prescribed)? Is it the case?

We now include the correlation plots for the Chem OFF simulation in Fig. 5. There is no significant correlation between ozone and dynamical heating rates in Chem OFF. We attribute this difference to the fact that there is no two-way interaction between ozone and model dynamics: a positive feedback before the onset of the negative feedback would allow the ozone concentration to be anomalously low under weak westerly conditions, which would enable the negative feedback to set in. We think that the timing between westerly background conditions and ozone forcing is crucial.

- Page 9, line 4-5/33-34. Please explain why you say that the final warming is more intense in Chem On. I would expect earlier final warmings to be "more dynamical", in the sense that the radiative forcing is still weak in late winter or early spring, and hence more abrupt and intense. And the opposite for late final warmings.

This is connected to the feedback we find in Chem ON. In the climatological mean we found a stronger PNJ right before an additional warming in Chem ON was detected due to differences in the dynamical heating rates that we find to be associated with the negative feedback in spring. We therefore conclude the break down of the polar vortex to be more abrupt in Chem ON as compared to Chem OFF. We did not consider early and late warmings yet. But it would be very interesting to consider in a future study.

**Technical comments:**

- Title: stratosphere-troposphere coupling?
Thank you for the remark. We adapted the title accordingly and removed the dash between "troposphere" and "coupling" when used in the manuscript.

- Page 9, line 21. "as well as".
corrected

[revised manuscript text omitted]